# NERDS: A General Framework to Train Camera Denoisers from Raw-RGB Noisy Image Paris

**Heewon Kim**[1] **Kyoung Mu Lee**[2,3]

[1] The Global School of Media, College of IT, Soongsil University, Korea
[2] ASRI, Department of ECE, Seoul National University, Korea
[3] IPAI, Seoul National University, Korea
`hwkim@ssu.ac.kr`  `kyoungmu@snu.ac.kr`

## Abstract

We aim to train accurate denoising networks for smartphone/digital cameras from raw-RGB noisy image pairs. Downscaling is commonly used as a practical denoiser for low-resolution images. Based on this processing, we found that the pixel variance of natural images is more robust to downscaling than the pixel variance of camera noise. Intuitively, downscaling removes high-frequency noise more easily than natural textures. To utilize this property, we can adopt noisy/clean image synthesis at low-resolution to train camera denoisers. On this basis, we propose a new solution pipeline – NERDS that estimates camera noise and synthesizes noisy-clean image pairs from only noisy images. In particular, it first models the noise in raw-sensor images as Poisson-Gaussian distributions, then estimates noise parameters using the difference of pixel variances by downscaling. We formulate the noise estimation as a gradient-descent-based optimization problem through a reparametrization trick. We further introduce a new Image Signal Processor (ISP) estimation method that enables denoiser training in a human-readable RGB space by transforming the downscaled raw images to the style of a given RGB noisy image. The noise and ISP estimations utilize rich augmentation to synthesize image pairs for denoiser training. Experiments show that NERDS can accurately train CNN-based denoisers (*e.g.*, DnCNN, ResNet-style network) outperforming previous noise-synthesis-based and self-supervision-based denoisers in real datasets.

## 1 Introduction

Image denoising is a conventional machine learning problem restoring original colors and patterns from noisy images. Deep-learning-based approaches have achieved breakthroughs in recent decades due to the power of neural networks. Early works (55; 38; 45) have successfully removed additive white Gaussian noise (AWGN), which allows network training under supervision by synthesizing noisy-clean image pairs.

Nevertheless, denoising images captured by smartphone/digital cameras poses an obstacle, as it is difficult to obtain clean images for noisy images with pixel-level alignment. Several works (2; 7) constructed datasets with the noisy-clean pairs for real-world images. Using these pairs (Figure 1(a)), many supervised-learning-based denoisers (51; 28; 52; 24; 15) restore crisp images on benchmarks from the datasets. However, constructing such datasets requires tightly controlled capturing environments, complicated post-processing, and massive human labor.

To overcome the drawback of plain supervised learning, two major types of research have been studied. The first line of works generates realistic noisy images from clean images to utilize supervised denoiser training as visualized in Figure 1(b). Several approaches (14; 11; 23; 26) adopt generative models using *unpaired* noisy-clean images based on GAN (20), but they achieve limited accuracy on real noise. Some other works synthesize realistic noise using existing noisy-clean image pairs (53; 1) or metadata for real cameras (6; 22), but they are limited in generalization for unseen noise. The second category aims to learn denoisers *without clean images*. The first work (34) in this category proposed the learning framework using multiple noisy images. After that, many self-supervised-learning approaches (5; 31; 10) use single noisy images (Figure 1(c)), which enable easy

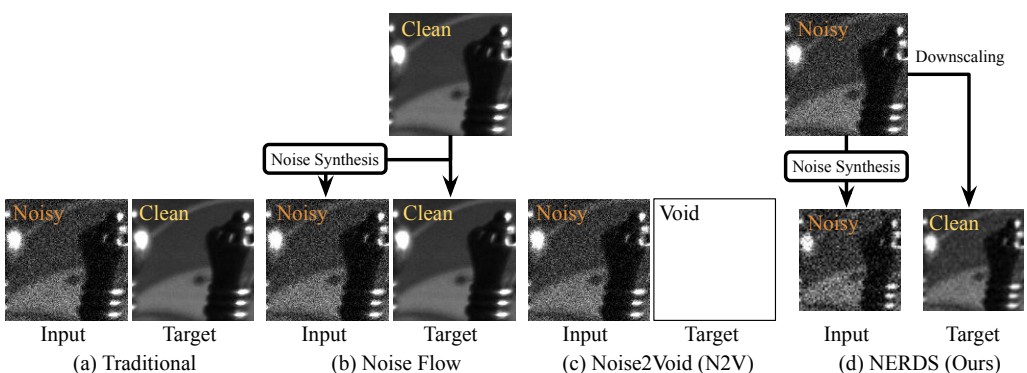

**Figure 1:** Different training schemes for CNN-based camera denoisers. (a) Traditionally, training denoisers requires pairs of noisy and clean images. However, clean target images are difficult to obtain from smartphone/digital cameras. (b) Noise Flow (1) generates realistic noise from clean images by learning real noise distributions using existing pairs of real images. (c) N2V (31) enables practical training from noisy images without clean targets but requires custom network architectures. (d) Our NERDS generates pseudo-noisy and pseudo-clean image pairs at low-resolution by utilizing image downscaling as a general denoiser and noise estimation through gradient-descent-based optimization.

data collection and denoiser adaptation to the test noise. However, they are still limited in real-world applications due to the requirements of custom network architectures and strong statistical noise assumptions.

To address the above limitations on camera denoiser training, we propose a new solution pipeline, namely Noise Estimation for RGB Denoising & Synthesis (NERDS), that generates noisy-clean image pairs from raw-RGB noisy image pairs. The pipeline composes three parts–noise estimation, ISP estimation, and denoiser training. We found that the pixel variance of natural clean images is robust to image downscaling, which is a widely-used denoiser for low-resolution images (41). The noise estimation adopts a Poisson-Gaussian noise model for raw images from sensors and optimizes its noise parameters by the pixel variances of downscaled images. The downscaled images and the estimated noise parameters enable generating pseudo-noisy and pseudo-clean image pairs at low-resolution (Figure 1(d)). Training denoisers on human-readable RGB images from real cameras has another issue: the conversion from the raw images to the RGB images is a black box. Our ISP estimation enables noise synthesis on the RGB space by learning RAW2RGB conversion using raw-RGB noisy image pairs[1]. Our denoiser training can utilize rich data augmentation based on estimated noise parameters and ISPs. Specifically, we introduce two techniques for this framework. First, a reparametrization trick allows estimating noise parameters through a gradient-descent-based optimizer. Second, a technique for style disentanglement from raw-RGB noisy image pairs. We summarize our contributions as follows:

- To the best of our knowledge, this is the first work to synthesize noisy-clean RGB image pairs at low-resolution for accurate camera denoiser training from raw-RGB noisy image pairs.

- We formulate noise estimation for Poisson-Gaussian noise as an optimization problem, and a novel reparameterization trick allows to estimate accurate noise parameters through gradient-descent.

- We propose a neural network that estimates the RAW2RGB conversions (or ISPs) used for given raw-RGB noisy image pairs. The ISP estimation generates realistic noisy-clean RGB image pairs from raw images.

- Our frameworks can train general CNN-based denoisers (*e.g.* DnCNN, ResNet-style network) accurately for given test noisy images by performing noise synthesis using them.

---

[1]Major camera manufacturers (*e.g.*, Samsung, Apple, Xiaomi, Cannon, and Sony) provide raw and RGB image pairs on their devices.

## 2 RELATED WORK

### 2.1 BLIND IMAGE DENOISING

**Traditional methods** Classical methods usually denoise noisy images without training data using wavelet (17), filtering (8) including BM3D (16), optimization (18; 37; 21), and effective prior (57). However, they perform limited accuracy compared to the recent deep-learning-based approaches.

**Supervised-learning-based methods** SIDD (2) and NIND (7) captured real noisy-clean image pairs to enable supervised-learning for real camera denoisers. However, the capturing procedure is unacceptably expensive and cumbersome. We describe the literature on realistic noise synthesis methods for the supervised-learning. DnCNN (55) introduced a neural network to remove additive white Gaussian noise (AWGN) for the first time. However, Guo *et al.* (22) demonstrated the limitation of AWGN denoisers to signal-dependent or spatially-correlated noise, which are known as the characteristics of real-world images. To alleviate this problem, two lines of works have been researched. The first category uses generative models for noise synthesis with stable learning (14), various noise characteristics (26), knowledge distillation (47), self-supervised-learning (11), and conditional adversarial networks (23). Nevertheless, the scene statistics mismatched between clean and noisy datasets make it difficult to train accurate denoisers in practice.

The second category investigates camera noise modeling. CBDNet (22) transforms AWGN into realistic noise by using signal-dependent noise parameters and simulating in-camera ISP functions, such as gamma correction and demosaicing. UPI (6) converts RGB images to raw images and simulates noise using metadata of specific cameras. CycleISP (53) trains neural networks for both RAW2RGB and RGB2RAW conversions on large-scale datasets with specific ISP and noise settings. Noise Flow (1) synthesizes raw noisy images using normalizing flow and metadata. The work on (12) proposed a GAN-based framework for noise generation in raw images which is adaptive to the camera. In (56), the authors claimed that the noise levels in metadata are inaccurate. Recently, the method in (30) models the RGB noise distribution using normalizing flow. SCUNet (54) proposed a practical noise model for a general-purpose denoiser.

However, all the above methods require real noisy-clean image pairs, ISPs, metadata, or high-quality clean images which are not always available. More importantly, the noise synthesis using predetermined training datasets leads to poor denoising performances on unseen noise. In contrast, the proposed method (NERDS) synthesizes realistic noisy-clean image pairs from only noisy images, which enables accurate denoiser training specialized in the test images.

**Self-supervised-learning-based methods** Noise2Noise (34) introduced a framework that trains denoisers using noisy images *only* for the first time. Noise2Void (31), Noise2Self (5), Noise2Same (48), and Neighbor2Neighbor (25) adopted advanced approaches which can train denoisers with single images corrupted by the i.i.d noise. Noisier2Noise (40), NAC (49), and R2R (42) add additive noise to the given noisy images to make auxiliary training pairs by using prior knowledge (or assumptions) of the noise distribution. Notably, the blind-spot network (BSN) in (31) has been improved by efficient architectures with small receptive fields (32) and dilated convolutions (47). Using the advanced BSNs, FBI-D (10) adopts a denoiser specialized in Poisson-Gaussian noise for real-world raw image denoising. AP-BSN (33) breaks the spatially-correlated noise of real-world RGB images by pixel-shuffle downsampling.

Although the above self-supervised-learning methods use practical training datasets (only noisy images), they require custom network architectures or strong noise assumptions. In contrast, NERDS performs noise synthesis that allows training general CNN-based denoisers based on the general noise modeling for smartphone/digital cameras.

### 2.2 NOISE ESTIMATION

Prior knowledge of noise distribution supports accurate restoration in most methods for image denoising, but it is not generally available in practice. Noise estimation methods alleviate this problem, especially for the additive white Gaussian noise (AWGN) and the Poisson-Gaussian noise. Principal component analysis (PCA) based approaches (35; 44; 13) perform accurate AWGN estimation. For the Poisson-Gaussian noise model, most existing methods (3; 46) including (19; 36) adopt two-step

approaches; estimating the local means and variances, then adopting maximum likelihood estimation (MLE) to fit the noise model. Foi *et al.*(19) proposed the Poisson-Gaussian noise model for the first time and a noise estimation algorithm based on wavelet decomposition. Liu *et al.* (36) adopted iterative patch selection for the generalized source-dependent noise. Recently, PGE-Net (10) adopted neural networks for accurate and fast optimization based on Generalized Anscombe Transformation (GAT) (4). In contrast, we proposed a noise estimation method for Poisson-Gaussian noise using natural scene statistics with gradient-descent-based optimization.

## 3 PRELIMINARY: NOISE MODELING

In a digital camera, an image sensor converts light into a digital signal (or a raw image), and an image signal processor (ISP) converts it into a human-readable RGB image. We regard that the noise of RGB images originates from the image sensor and is transformed by the ISP. Thus, we model noise distribution of raw images and RAW2RGB conversion including ISPs.

**Raw Image: Poisson-Gaussian (P-G) Noise**   A common noise model for raw images follows the Poisson-Gaussian distribution (19), defined as:

$$\boldsymbol{x} \sim \mathcal{N}(\boldsymbol{z}, \beta_1^2 \boldsymbol{z} + \beta_2^2), \tag{1}$$

which is a heteroscedastic Gaussian where $\boldsymbol{z}$ is the true signal, $\boldsymbol{x}$ is a raw noisy image observed on real image sensors, and $\beta_1$, $\beta_2 \geq 0$ are signal-dependent and signal-independent noise parameters. Real cameras provide $\boldsymbol{x}$ as well as $\beta_1$ and $\beta_2$ in metadata. However, the noise parameters in metadata are often inaccurate (56). Section A.1.1 discusses the noise distribution of raw images and the noise parameters in metadata further.

**RGB Image: Transformed Noise**   ISPs transform raw images into RGB images using nonlinear functions, such as demosaicing, white balancing, color correction, and tone mapping. The transformation changes noise distribution (e.g., breaking the i.i.d property in equation 1). Moreover, users can retouch image tones and colors, altering the noise distribution further. We define such RAW2RGB conversion ($\boldsymbol{T}$) as a function of the image style ($\boldsymbol{s}$),

$$\boldsymbol{y} \equiv T(\boldsymbol{x}; \boldsymbol{s}), \tag{2}$$

where $\boldsymbol{x}$ is a raw noisy image obtained from real image sensors and $\boldsymbol{y}$ is a RGB noisy image with custom tones and colors. The real noise datasets (2; 43) used a simple and open-source ISP for the entire dataset without any image retouching processes. However, modern smartphones and digital cameras use custom ISPs with hidden internal functions. They provide multiple image styles depending on scenes.

The following section provides a general framework to estimate P-G noise parameters ($\beta_1$, $\beta_2$) and RAW2RGB conversion ($\boldsymbol{T}$) with image style ($\boldsymbol{s}$) using only noisy images ($\boldsymbol{x}$, $\boldsymbol{y}$).

## 4 PROPOSED METHOD

### 4.1 OVERVIEW

For a given noisy pair of raw ($\boldsymbol{x}$) and RGB ($\boldsymbol{y}$) images, the proposed method composes three steps to restore the RGB clean image without auxiliary training data. First, we estimate noise parameters ($\beta_1$, $\beta_2$) from $\boldsymbol{x}$ (Section 4.3). Second, we learn the conversion from $\boldsymbol{x}$ to $\boldsymbol{y}$ while disentangling image style ($\boldsymbol{s}$) (Section 4.4). Third, we synthesize diverse pairs of noisy-clean images to train arbitrary RGB denoisers with a general supervised-learning framework (Section 4.5).

### 4.2 OBSERVATION

We first illustrate our observation on raw noisy images with downscaling. Specifically, we investigate the statistical characteristics variances of 256×256 images in the validation set of SIDD (2). To evaluate their noise levels, we downscale all raw images for each color channel with the scaling factor of 2, and rank these images according to the differences of pixel variances before and after

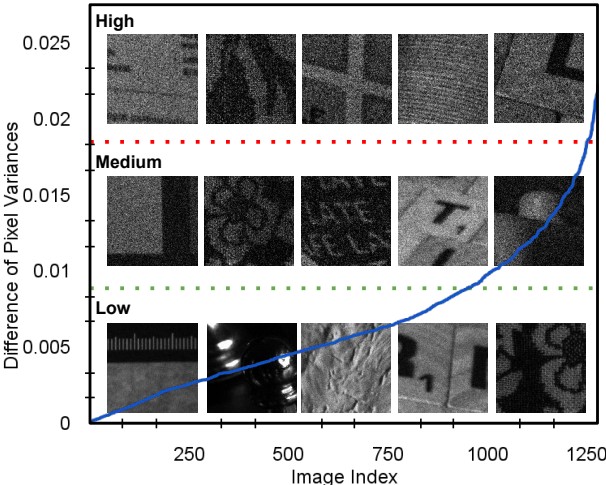

**Figure 2:** The ranked curve for the difference of pixel variances through downscaling on images from SIDD validation and the visualization of noise levels.

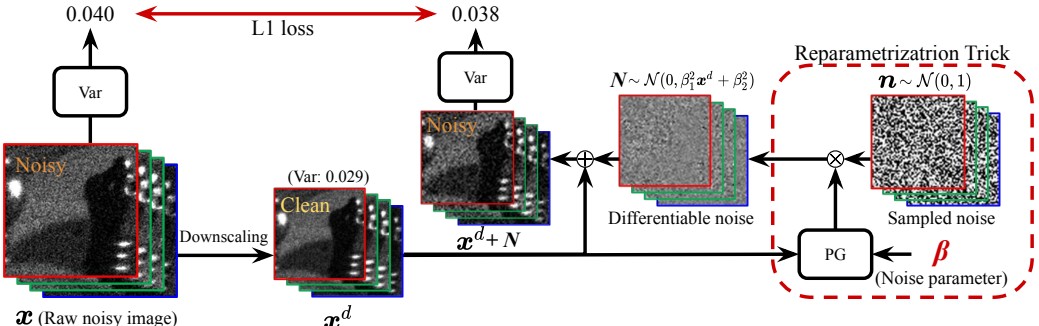

**Figure 3:** Noise estimation from a raw noisy image with a reparametrization trick.

downscaling. As visualized in Figure 2, we show these values in a blue curve and separate them into three levels –low, medium, high. It is observed that the images with high differences are very noisy, while the images with low differences present clean textures. This phenomenon indicates that the true signal $\boldsymbol{z}$ is more robust to the pixel variances through downscaling than real noise. That is why we propose the following method, which uses the difference of pixel variances to estimate noise levels without clean images.

### 4.3 POISSON-GAUSSIAN NOISE ESTIMATION VIA A REPARAMETERIZATION TRICK

We aim to estimate noise for $\boldsymbol{x}$ without any information other than $\boldsymbol{x}$. For this, we find an additive noise ($\boldsymbol{N}$) for the downscaled image ($\boldsymbol{x}^d$) by solving the following optimization problem,

$$\min_{\boldsymbol{N}} |Var(\boldsymbol{x}) - Var(\boldsymbol{x}^d + \boldsymbol{N})|, \tag{3}$$

where $\boldsymbol{N} \sim \mathcal{N}(0, \beta_1^2 \boldsymbol{x}^d + \beta_2^2)$ is the Poisson-Gaussian noise discussed in equation 1 and $Var(\cdot)$ denotes a function that outputs the pixel variance of the input image. We can approximate $\beta_1$ and $\beta_2$ as noise parameters for $\boldsymbol{x}$, given that the difference of pixel variances through downscaling is correlated with noise levels, as discussed in Section 4.2. Section A.1.2 analyzes the downscaling effect further. However, equation 3 is difficult to optimize due to the non-differentiable process of noise sampling. To alleviate this problem, we introduce a reparameterization trick that separates noise sampling into learnable parameters and sampling from a normal distribution. Formally, we reformulate equation 3 as

$$\min_{\boldsymbol{\beta}} |Var(\boldsymbol{x}) - Var(\boldsymbol{x}^d + PG(\boldsymbol{\beta}, \boldsymbol{x}^d) \times \boldsymbol{n})|, \tag{4}$$

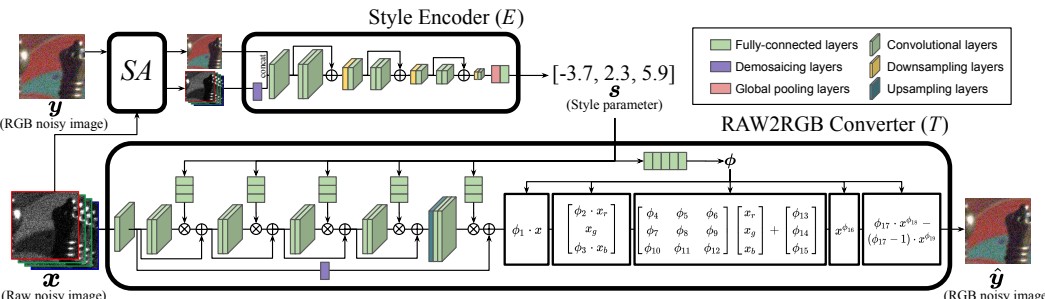

**Figure 4:** ISP estimation from a pair of raw-RGB noisy images with style disentanglement.

where $PG(\boldsymbol{\beta}, \boldsymbol{x}^d) = (\beta_1^2 \boldsymbol{x}^d + \beta_2^2)^{0.5}$ denotes a Poisson-Gaussian converter and $\boldsymbol{n} \sim \mathcal{N}(0, 1)$ is additive noise with a normal distribution. Given that the only learnable parameters in equation 4 are $\beta_1$ and $\beta_2$, a gradient-descent optimizer (*e.g.*, ADAM (29)) easily finds the noise parameters. Figure 3 visualizes the optimization problem used as noise estimation for $\boldsymbol{x}$.

We are surprised that such a simple optimization problem estimates accurate noise parameters, while downscaled images can contain remaining noise or over-smoothing textures. This result is because the well-designed optimizer learns the characteristics of signal-dependent and signal-independent noise over the entire image content.

So far we have discussed noise estimation and synthesis on raw images. For more practical applications, the following section describes RGB noise synthesis by estimating RAW2RGB conversion.

### 4.4 ISP Estimation with Style Disentanglement

We aim to learn a RAW2RGB conversion (or an ISP) from $\boldsymbol{x}$ to $\boldsymbol{y}$. However, a naïve network training from $\boldsymbol{x}$ to $\boldsymbol{y}$ can easily overfit to the image contents. Moreover, to generate multiple styles of RGB images, training a neural network for each style is resource intensive. For this, we disentangle the styles from RGB images, allowing a network to learn multiple-style generation.

Specifically, our ISP estimation composes two networks, the style encoder ($E$) and the RAW2RGB converter ($T$) (Figure 4). $E$ identifies the image style as a style parameter ($\boldsymbol{s}$) from the raw noisy image ($\boldsymbol{x}$) and the RGB noisy image ($\boldsymbol{y}$) while $T$ uses $\boldsymbol{x}$ and $\boldsymbol{s}$ as input to generate an RGB image ($\hat{\boldsymbol{y}}$). Then, we can recall the equation equation 2 as

$$\hat{\boldsymbol{y}} = T(\boldsymbol{x}; E(\boldsymbol{x}, \boldsymbol{y})), \tag{5}$$

where $\boldsymbol{s} = E(\boldsymbol{x}, \boldsymbol{y})$ while training $E$ and $T$ to minimize L1 differences between $\hat{\boldsymbol{y}}$ and $\boldsymbol{y}$. $E$ composes 6 residual blocks, global pooling, and a fully connected layer and $T$ composes 4 residual blocks and parameterized ISP functions (27). We use the residual blocks of two convolutional layers and one ReLU activation with 64 filters and 3×3 kernels. The training data can compose a single pair or multiple pairs of raw-RGB noisy images. For accurate RAW2RGB conversion for downscaled images, which are unseen in training, we first adopt image scale augmentation ($SA$) that changes image resolution. $SA$ generates multiple training patches for a style (or a patch) by randomly sampled scaling factors to prevent overfitting to a few training data. Formally, we redefine the style parameter ($\boldsymbol{s}$) as follows,

$$\boldsymbol{s} \equiv E(SA(\boldsymbol{x}, \boldsymbol{y})). \tag{6}$$

Second, we design a bottleneck structure that lowers the dimension of $\boldsymbol{s}$ to avoid encoding of information about image contents. $E$ reduces channels and resolutions of $\boldsymbol{s}$ ($\boldsymbol{s} \in \mathbb{R}^{3 \times 1 \times 1}$ in this paper). $\boldsymbol{s}$ conditions $T$ by channel attention and ISP parameter generation via fully connected layers.

### 4.5 Denoiser Training with Data Augmentation

We synthesize noisy-clean RGB image pairs at low-resolution to train general RGB denoisers. The pseudo-noisy/pseudo-clean images are the downscaled raw image ($\boldsymbol{x}^d$) with/without additive noise $\boldsymbol{N}$ transformed to RGB space using $T$. Formally, the denoiser ($D$) training minimizes the following objective function,

$$\min_D |T(\boldsymbol{x}^d; \boldsymbol{s}) - D(T(\boldsymbol{x}^d + \boldsymbol{N}; \boldsymbol{s}))|, \tag{7}$$

where $s$ is the style parameter in equation 6. Our denoising framework allows rich augmentation for data synthesis. First, scaling and intensity augmentation $(SIA)$ increases content diversity by changing the image resolution and pixel values from $0.5\times$ to $1.5\times$. Second, we randomly scale noise parameters, $\beta_1$ and $\beta_2$, from $0.5\times$ to $1.5\times$. This augmentation alleviates the noise estimation error in Section 4.3. Lastly, when multiple noisy images are given, we can augment the noise parameters and the style parameters across different images. Overall, the objective function of our denoiser training equation 7 becomes

$$\min_D |T(SIA(\boldsymbol{x}^d); \boldsymbol{s}') - D(T(SIA(\boldsymbol{x}^d) + \boldsymbol{N}'; \boldsymbol{s}'))|, \qquad (8)$$

where $SIA$ is the scale and intensity augmentation, $\boldsymbol{s}'$ is the augmented style parameter, and $\boldsymbol{N}'$ is the additive noise with augmented noise parameters. At testing, the denoiser takes the RGB noisy image ($\boldsymbol{y}$) as input, just like conventional CNN-based approaches.

## 5 EXPERIMENTS

### 5.1 IMPLEMENTATION DETAILS

**Downscaling** We downscale a raw image $\boldsymbol{x}$ to $\boldsymbol{x}^d$ by bicubic interpolation after asymmetric 2D Gaussian blurring with the kernel size of $21 \times 21$. We randomly select the standard deviation of the Gaussian blur from $0.25 \times ds$ to $0.75 \times ds$ for each dimension, where $ds$ denotes downscaling factor randomly selected in the range of $[1.5, 2.5]$. We use the same hyper-parameters for noise estimation, ISP estimation, and denoiser training unless otherwise specified.

**Optimization** We use ADAM (29), $128 \times 128$ image patches, and a batch size of 64 for all experiments. Noise estimation adopts $2.5 \times 10^4$ iterations with the initial learning rate of $1 \times 10^{-4}$ which becomes a tenth part in every $5 \times 10^3$ iterations. We use the same initial learning rates for ISP estimation and denoiser training for $5 \times 10^5$ iterations without the learning rate decay.

**Denoiser** We use two simple networks as denoiser to present the generalizability of the proposed training scheme. Specifically, NERDS+DnCNN uses DnCNN (55) and NERDS+D uses a ResNet-style architecture, composing a global skip connection and 32 residual blocks. Each block has two convolutional layers and one ReLU activation with 64 filters and $3 \times 3$ kernels.

**Dataset** We use BSD68 (39) which consists of 68 gray-scale images to evaluate the performance of noise estimation. For real image denoising, we use raw noisy and RGB noisy images on SIDD (2), DND (43), and MIT-Adobe FiveK (9). SIDD (2) consists of training, validation, and benchmark datasets. We use 1,280 $256 \times 256$ patches from 40 noisy images on the benchmark for denoiser training. In the setting of extra images, we use 50 noisy images on the training dataset with low ISO levels of 100 and downscale the noisy images with the scaling factor from 1.1 to 1.2. For DND (43), we use 1,000 $512 \times 512$ patches from 50 noisy images on the benchmark. We evaluate the denoising performances on the benchmarks by submitting the results to the public websites for SIDD and DND. MIT-Adobe FiveK (9) consists of 5,000 raw images and paired RGB images retouched by 5 photographers. Each RGB image has its own RAW2RGB conversion using Adobe Lightroom while the raw image contains real camera noise. We demonstrate an extreme scenario where only 5 noisy images are available for denoiser training.

### 5.2 RESULTS FOR NOISE ESTIMATION AND SYNTHESIS

We first validate our noise estimation on additive Poisson-Gaussian noise to images on BSD68 (39), and then visualize the noise synthesis at low-resolution from noisy images on MIT-Adobe FiveK (9).

**Poisson-Gaussian noise estimation** We demonstrate the effectiveness of our NERDS-raw on P-G noise estimation by comparing with Foi *et al.* (19), Liu *et al.* (36), and PGE-Net (10) which estimate P-G noise parameters from noisy images. Table 1 presents the average of the noise parameters estimated from each image on BSD68 with four different noise levels. In most cases, NERDS-raw estimates the most accurate noise parameters.

**Table 1:** Performance comparison of Poisson-Gaussian noise estimation. The reported scores are average values of $(\hat{\beta}_1, \hat{\beta}_2)$ estimated from BSD68 with additive Poisson-Gaussian noise level of $(\beta_1, \beta_2)$. Bold denotes the best result.

| Noise level $(\beta_1, \beta_2)$ | Foi *et al.* $(\hat{\beta}_1, \hat{\beta}_2)$ | Liu *et al.* $(\hat{\beta}_1, \hat{\beta}_2)$ | PGE-Net $(\hat{\beta}_1, \hat{\beta}_2)$ | NERDS-raw (Ours) $(\hat{\beta}_1, \hat{\beta}_2)$ |
|---|---|---|---|---|
| (0.100, 0.0200) | (0.096, 0.042) | (0.072, 0.045) | (0.098, 0.0030) | (**0.100**, **0.0229**) |
| (0.100, 0.0002) | (0.097, 0.035) | (0.071, 0.044) | (0.095, **0.0001**) | (**0.101**, **0.0001**) |
| (0.050, 0.0200) | (0.049, 0.031) | (0.040, 0.040) | (0.052, 0.0001) | (**0.051**, **0.0245**) |
| (0.050, 0.0002) | (**0.051**, 0.018) | (0.039, 0.034) | (**0.051**, 0.0001) | (0.052, **0.0002**) |

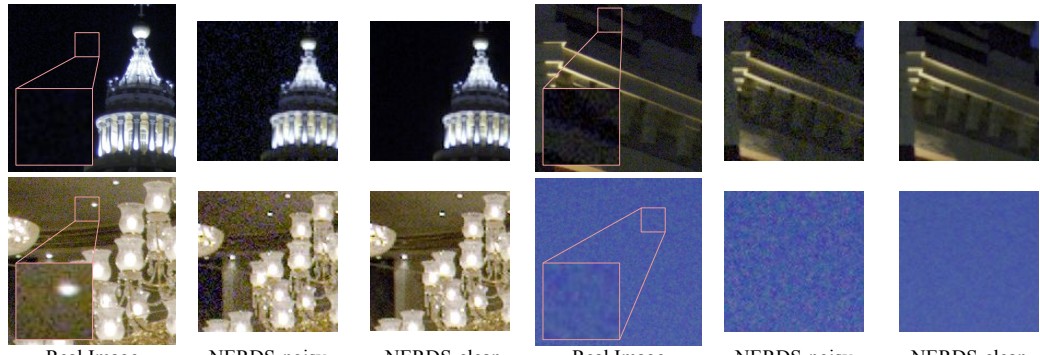

| Real Image | NERDS-noisy | NERDS-clean | Real Image | NERDS-noisy | NERDS-clean |

**Figure 5:** Noise synthesis results using NERDS. NERDS can synthesize noisy-clean RGB image pairs at low-resolution from raw-RGB noisy image pairs. NERDS-noisy and NERDS-clean denote synthetic RGB images with and without noise. This example uses the downscaling factor of 2. (*Zooming-in for the best view.*)

**Noisy-clean RGB image synthesis**    Figure 5 shows noise synthesis results using NERDS. Each real image from MIT-Adobe FiveK has its own RAW2RGB conversion. NERDS-noisy contains realistic noise specialized to each real image, such as blue/dark noise in dark areas with none i.i.d characteristics. Figure 9 visualizes more pseudo-noisy and pseudo-clean image pairs on SIDD/DND datasets. To the best of our knowledge, NERDS is the first work to generate the realistic noisy images and the paired clean images for the unknown RAW2RGB conversions (or ISPs) by using only raw-RGB noisy image pairs. Existing works for noise synthesis (6; 54; 53; 1; 30) require real noisy-clean image pairs, metadata, and ISPs. Thus, naïve comparisons between NERDS and the existing methods are unfair, but we present them in Section A.3.1.

### 5.3    COMPARISONS TO THE SOTA DENOISING METHODS

**Results on benchmarks**    Given that NERDS trains denoisers without clean images, we compare our denoisers with the SotA denoising methods that *do not use the pairs of real noisy-clean images*. Specifically, we compare GAT+BM3D (16), N2V (31), AP-BSN (33), FBI-D (10), C2N+DIDN (26), and SCUNet (54) on SIDD and DND benchmarks. Table 2 presents the effectiveness of the proposed denoisers, NERDS+DnCNN and NERDS+D. C2N+DIDN (26) synthesizes realistic noise using *unpaired* noisy-clean images, while our NERDS+DnCNN outperforms it with the simpler denoiser (DIDN *vs.* DnCNN). SCUNet (54) synthesizes realistic noise using high-quality clean images and the predetermined noise models including specific noise levels and ISP pipelines. The other works require only noisy images via prior-based filtering (16), self-supervised learning (31; 33; 47), and noise estimation (10). The works without clean images perform denoising in raw image space and received the results in RGB image space by submitting the denoised raw images to the public websites (2; 43). They often fail to remove noise in RGB image space, given that they assume strong noise characteristics such as Poisson-Gaussian distributions and the i.i.d property that do not hold in the RGB image space. Section A.2 presents a generalization test for NERDS+D with latency analysis to denoise a test image from scratch. Section A.3.2 provides the comparisons with the noise synthesis methods which use the pairs of real noisy-clean images. Figure 10, 11, and 12 visualize more denoising results on SIDD, DND, and MIT-Adobe FiveK.

**Ablation Study**    Our NERDS enables rich and effective augmentation for denoiser training. Table 3 demonstrates the ablation study on the data augmentation. The setting without image scale augmentation (Table 3(1)) uses a fixed downscaling factor of 2. Each component improves the

**Table 2:** Performance comparison with SotA denoising methods. The reported scores are PSNR (dB)/SSIM on RGB images from the SIDD and DND benchmarks. Bold denotes the best result. ✓denotes accessible types of images at training. Extra images denote images other than the noisy images of the benchmark.

| Dataset | GAT+BM3D | N2V | AP-BSN | FBI-D | C2N+DIDN | SCUNet | NERDS+DnCNN | NERDS+D | |
|---|---|---|---|---|---|---|---|---|---|
| Clean images | - | - | - | - | ✓ | ✓ | - | - | - |
| Extra images | - | - | - | ✓ | ✓ | ✓ | - | - | ✓ |
| Synthetic pairs | - | - | - | - | ✓ | ✓ | ✓ | ✓ | ✓ |
| SIDD | 34.61/0.879 | 32.85/0.847 | 36.91/0.931 | 38.07/0.942 | 35.35/0.937 | 22.89/0.797 | 36.42/0.923 | 37.40/0.941 | **38.28/0.949** |
| DND | 37.98/0.920 | 35.82/0.902 | 38.09/0.937 | 38.98/0.945 | 37.28/0.924 | - | 38.21/0.941 | **39.34/0.950** | - |

**Table 3:** Ablation study on data augmentation to train NERDS+D. PSNR (dB)/SSIM on RGB images from SIDD validation.

| Augmentation | (1) | (2) | (3) | (4) | (5) | (6) |
|---|---|---|---|---|---|---|
| Image scale | - | ✓ | ✓ | ✓ | ✓ | ✓ |
| Image intensity | - | - | ✓ | ✓ | ✓ | ✓ |
| Noise parameter | - | - | - | ✓ | ✓ | ✓ |
| Style parameter | - | - | - | - | ✓ | ✓ |
| Extra images | - | - | - | - | - | ✓ |
| SIDD | 35.63/0.897 | 36.79/0.928 | 37.16/0.935 | 37.84/0.944 | 38.02/0.946 | **38.51/0.950** |

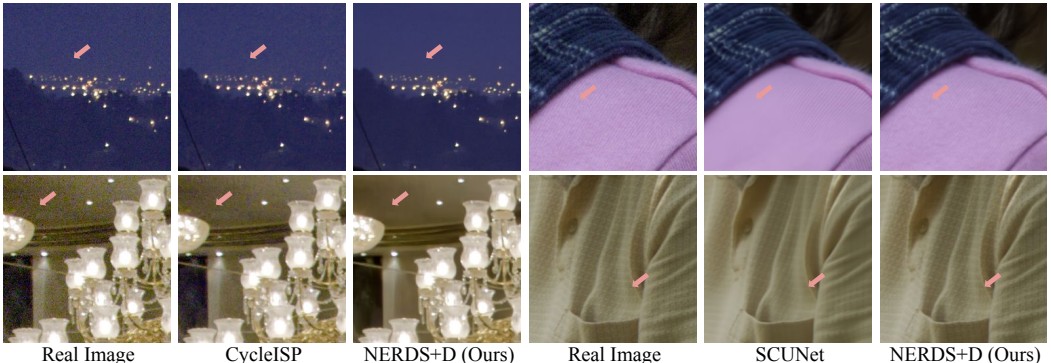

|  |  |  |  |  |  |
|---|---|---|---|---|---|
| Real Image | CycleISP | NERDS+D (Ours) | Real Image | SCUNet | NERDS+D (Ours) |

**Figure 6:** Qualitative comparisons of denoising results on MIT-Adobe FiveK (9). Pink arrows indicate the remaining noise of CycleISP and the over-smooth textures of SCUNet while our denoiser generates crisp images. (*Zooming-in for the best view.*)

restoration performances. In particular, noise parameter augmentation improves over than 0.6 dB, given that it can alleviate noise estimation error. The usage of extra images enables high-quality clean image synthesis by downscaling images with low noise levels and small scaling factors.

**Results on MIT-Above FiveK** We evaluate CycleISP (53), SCUNet (54), and NERDS+D on retouched images from MIT-Adobe FiveK (9). CycleISP (53) employs supervised learning on SIDD training dataset with additional synthetic data using the predetermined noise levels and ISP pipeline. Nevertheless, CycleISP fails to denoise the retouched image as visualized in Figure 6. SCUNet (54) is trained on high-quality images with practically designed additive noise, but SCUNet often generates over-smooth textures. These results are due to different noise distributions between the training images and the test images. In contrast, NERDS+D removes severe real noise while maintaining image details. We report more qualitative results in Figure 12.

# 6 CONCLUSION

We present a general framework to train denoisers from noisy images, called NERDS. The framework composes noise estimation, ISP estimation, and denoiser training. For noise synthesis, we estimate Poisson-Gaussian noise in raw images and ISP (or RAW2RGB conversion) for each RGB image. NERDS allows rich data augmentation for accurate denoiser training. Experimental results show the state-of-the-art restoration accuracy on real noise benchmarks.

**Acknowledgements.** This work was supported in part by the IITP grants [No.2021-0-01343, Artificial Intelligence Graduate School Program (Seoul National University), No.2022-0-00156, No. 2021-0-02068, and No.2022-0-00156], and the NRF grant [No. 2021M3A9E4080782] funded by the Korea government (MSIT), and AIRS Company in Hyundai Motor Company & Kia Corporation through HMC/KIA-SNU AI Consortium Fund.

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

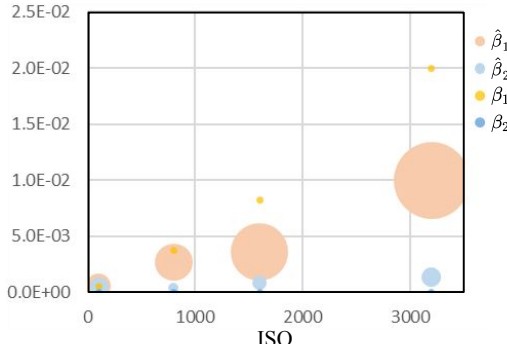

**Figure 7:** Noise distribution of images from Galaxy S6. $\beta_1$ and $\beta_2$ are the noise levels of metadata, while $\hat{\beta}_1$ and $\hat{\beta}_2$ are the estimated noise parameters for Poisson-Gaussian distribution using NERDS. While $(\beta_1, \beta_2)$ have specific values at each ISO level, the radius of the circles for $\hat{\beta}_1$ and $\hat{\beta}_2$ represents the standard deviation of the estimated values for each image on SIDD training dataset.

**Table 4:** Ablation study of P-G noise parameters to train denoisers. Each method provides different values of noise parameters for the same denoiser architecture, augmentation techniques, and training schemes. The reported scores indicate restoration performances on RGB images from the SIDD validation. NERDS-raw achieves the best PSNR score.

|  | Metadata | Random | NERDS-raw (Ours) |
|---|---|---|---|
| PSNR (dB)/SSIM | 37.56/0.937 | 37.92/0.941 | **38.51/0.950** |

# A APPENDIX

## A.1 DISCUSSIONS

### A.1.1 NOISE DISTRIBUTION OF RAW IMAGES

Our NERDS assumes that the raw images have the noise that follows Poisson-Gaussian distribution. However, the image sensor of a smartphone or a digital camera is a black box. We do not know what kind of post-processing has been applied to the raw images, or whether the noise level (parameters) in the metadata represent the proper parameters of Poisson-Gaussian distribution. For instance, one of the noise levels in the metadata of Galaxy S6 ($\beta_2$ in Figure 7) equals zero for all images, which is theoretically impossible for both shot and read noise parameters of the Poisson-Gaussian model. Nonetheless, Figure 7 presents a similar tendency between the noise levels in the metadata ($\beta_1$, $\beta_2$) and the estimated noise parameters ($\hat{\beta}_1$, $\hat{\beta}_2$). Thus, we approximate the noise distribution in raw images as Poisson-Gaussian noise and show that NERDS+D can achieve satisfactory denoising results using the estimated Poisson-Gaussian noise.

To show the effectiveness of accurate P-G noise parameter estimation, Table 4 presents an ablation study of the values of P-G noise parameters. NERDS-raw, Metadata, and Random represent the restoration performance of an RGB image denoiser trained with the noise parameters from each method. NERDS-raw indicates the model in Table 3(6), Metadata uses the noise levels from the metadata, and Random uniformly samples noise parameters between the maximum and minimum values in NERDS-raw. Metadata and Random use the same denoiser architecture (D), augmentation techniques, and training schemes with NERDS-raw except the values of noise parameters for a fair comparison.

### A.1.2 CLEAN IMAGE VIA DOWNSCALING

The *optimal* clean image via downscaling is a noise-free low-resolution image that has the same statistics as the true signal (clean high-resolution image). Empirically, we regard the raw images downscaled after burring (low-pass filtering) as raw *pseudo*-clean images. However, blurring images breaks the optimal setting when image structures are too small (*e.g.*, 1-pixel dots) or the noise is too severe compared to the size of the blur. Figure 8 shows the evidence and limitations of the utility of pseudo-clean images used in NERDS. First, the images downscaled without pre-blurring have

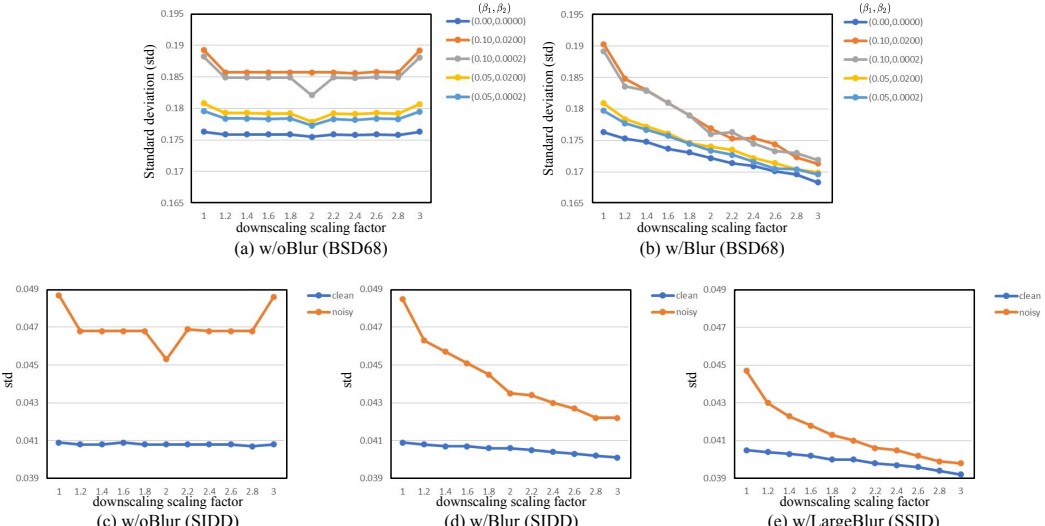

**Figure 8:** Standard deviation (std) of downscaled images. Blue lines denote the std of clean images while the others denote the std of noisy images. Downscaling without pre-blurring maintains the std values through scaling on both BSD68 ((a)) and SIDD ((c)). In contrast, blurring before downscaling reduces the std values of noisy images drastically than clean images ((b) and (d)). (e) Large blur kernels ($2\times$ larger than the kernels used in NERDS) reduce the std values more steeply.

**Table 5:** $\mathcal{D}_{KL}$ comparison for ablation study of NERDS-raw with the different sizes of blur kernels. NERDS-raw w/Blur, which is the original NERDS-raw, achieves the best performance.

| NERDS-raw | w/oBlur | w/Blur (Ours) | w/LargeBlur |
|---|---|---|---|
| $\mathcal{D}_{KL}$ | 0.1710 | **0.0344** | 0.0535 |

similar statistics to those of the images before downscaling. This indicates that the image scaling can be regarded as capturing an image at different distances between an object and a camera. Second, the blurring operation reduces the std values of noisy images drastically more than clean images. This is why we use downscaled images as pseudo-clean images. However, blurring also reduces the std values of clean images, making it challenging to find *optimal* clean images. To alleviate this difficulty, we estimate the noise by gradient-descent-based optimization and adopt augmentation of noise parameter scaling ($\times 0.5 \sim \times 1.5$) for accurate denoiser training.

We further analyze the effectiveness of blur kernels by measuring KL divergence between synthesized noisy images and real noisy images. The lower value, the better. Table 5 visualizes an ablation study on the size of blur kernels on the SIDD dataset. NERDS-raw w/Blur achieves the best performance compared to both cases of w/oBlur and w/LargeBlur. These results indicate that the downscaled images work as clean images well with the proper size of blur kernels.

## A.2 EFFICIENT INFERENCE AT TEST TIME

The proposed method composes three steps (noise estimation, ISP estimation, and denoiser training) and an additional step of denoiser testing. Each step described above has the latency described in Table 6. We use GeForce RTX 2080 Ti GPU and an HD image for testing.

The noise estimation, ISP estimation, and denoiser training using noisy test images are time-consuming. To skip the processes at test time, Table 7 presents a generalization test for NERDS+D trained on different datasets. NERDS+D trained on DND has accurate restoration performance on DND but performs 2 dB lower PSNR on SIDD than the model trained on SIDD. In contrast, NERDS+D trained on SIDD performs accurate restoration on both datasets. This phenomenon indicates that SIDD contains noise distributions similar to DND and that well-designed noisy images enable generalized denoiser training. For instance, camera manufacturers can collect training datasets of only noisy images concerning the image sensor, ISP, and expected image retouching.

**Table 6:** Latency analysis of the processes for NERDS and NERDS+D.

| | Noise Estimation | ISP Estimation | Denoiser Training | Denoiser Testing |
|---|---|---|---|---|
| Latency | 30 m | 1 h | 6 h | 0.1 s |

**Table 7:** Generalization test for NERDS+D on different training datasets without clean images. The reported scores are PSNR (dB)/SSIM on RGB images from the SIDD validation and the DND benchmark. Bold denotes the best result.

| NERDS+D | | Taining | |
|---|---|---|---|
| | | SIDD | DND |
| Testing | SIDD | **38.51/0.950** | 36.26/0.923 |
| | DND | 39.14/0.949 | **39.34/0.950** |

### A.3 COMPARISONS TO NOISE SYNTHESIS METHODS USING REAL IMAGE PAIRS

The recently proposed methods (1; 12; 56; 30) synthesize noisy images with novel noise models. These methods use noise/clean image pairs and metadata that are not accessible at test time, such as the DND dataset. Although the proposed method is a general framework for denoising training without a clean image, we present comparisons with these methods by evaluating the noise estimation results with denoising performance and KL divergence.

**Table 8:** $\mathcal{D}_{KL}$ comparison with SotA noise synthesis methods. NERDS-raw w/clean image outperforms all compared methods.

| | Calibrated P-G (56) | Noise Flow (1) | Camera-Aware (12) | NERDS-raw (Ours) | |
|---|---|---|---|---|---|
| Clean data | ✓ | ✓ | ✓ | ✓ | |
| $\mathcal{D}_{KL}$ | 1.5147 | 0.0481 | 0.0144 | **0.0079** | 0.0344 |

**Table 9:** Raw image denoising comparison with SotA noise synthesis methods on the SIDD benchmark. NERDS-raw achieves the best PSNR score. We use DnCNN as a denoiser for NERDS-raw.

| | Gaussian | Noise Flow (1) | Camera-Aware (12) | NERDS-raw (Ours) |
|---|---|---|---|---|
| PSNR (dB)/SSIM | 43.63/0.968 | 48.52/0.992 | 48.71/**0.993** | **48.93**/0.985 |

**Table 10:** RGB image denoising comparison with SotA noise synthesis methods on the SIDD benchmark. NERDS achieves the best PSNR/SSIM scores. All methods use DnCNN as the denoiser.

| | Gaussian | Noise Flow (1) | C2N (26) | RGB Noise Flow (30) | NERDS (Ours) |
|---|---|---|---|---|---|
| PSNR (dB)/SSIM | 32.72/0.873 | 33.81/0.894 | 33.76/0.901 | 34.74/0.912 | **36.42/0.923** |

#### A.3.1 RESULTS FOR NOISE ESTIMATION AND SYNTHESIS

Table 8 presents the comparisons of KL divergence ($\mathcal{D}_{KL}$) between synthesized noisy images and real noisy images. The lower values, the better. The proposed method outperforms all compared methods when using clean images. We could not reproduce the results of RGB Noise Flow (30) since the source code was not available. The score reported in the paper is 0.044 for RGB Noise Flow, where Noise Flow scores 0.198. The scores of KL divergence are dependent on hyperparameters.

#### A.3.2 RESULTS FOR RAW/RGB IMAGE DENOISING

Table 9 and 10 present denoising performances for raw/RGB images on the SIDD benchmark, where NERDS-raw and all methods for RGB images use DnCNN as a denoiser. Although NERDS-raw and NERDS do not use clean images for noise estimation, noise synthesis, and denoiser training, NERDS-raw and NERDS achieve the best PSNR scores in each table. When converting the denoised raw images using NERDS to RGB images, the PSNR is 35.74 dB which is lower than RGB image denoising (36.42 dB).

### A.4 ADDITIONAL ABLATION STUDY

**Ablation Study for Noise Estimation (NERDS-raw).** For noise estimation (NERDS-raw), we design ablation studies on blurring strengths and downscaling factors on BSD68 in Table 11 and 12.

**Table 11:** Ablation study for noise estimation on blurring strengths.

| Noise level $(\beta_1, \beta_2)$ | w/oBlur $(\hat{\beta}_1, \hat{\beta}_2)$ | w/Blur (Ours) $(\hat{\beta}_1, \hat{\beta}_2)$ | w/LargeBlur $(\hat{\beta}_1, \hat{\beta}_2)$ |
|---|---|---|---|
| (0.100, 0.0200) | (0.080, 0.144) | (**0.100, 0.0229**) | (0.119, 0.0220) |

**Table 12:** Ablation study for noise estimation on downscaling factors (DF).

| Noise level $(\beta_1, \beta_2)$ | DF 1 $(\hat{\beta}_1, \hat{\beta}_2)$ | DF 1.5~2.5 (Ours) $(\hat{\beta}_1, \hat{\beta}_2)$ | DF 2.5~4.5 $(\hat{\beta}_1, \hat{\beta}_2)$ |
|---|---|---|---|
| (0.100, 0.0200) | (0.065, 0.0085) | (**0.100, 0.0229**) | (**0.100**, 0.0279) |

**Table 13:** Ablation study for denoiser training on blurring strengths.

| | w/oBlur | w/Blur (Ours) | w/LargeBlur |
|---|---|---|---|
| PSNR (dB)/SSIM | 26.99/0.642 | **38.02/0.946** | 38.01/0.944 |

**Table 14:** Ablation study for denoiser training on blurring strengths.

| | DF 1 | DF 1.5~2.5 (Ours) | DF 2.5~4.5 |
|---|---|---|---|
| PSNR (dB)/SSIM | 37.66/0.942 | **38.02/0.946** | 37.89/0.943 |

The setting of w/LargeBlur uses two times larger blur kernels than NERDS-raw. The settings of w/Blur and DF 1.5~2.5, which indicates NERDS-raw in Table 1, present better performance than compared settings. The settings of w/oBlur or DF 1 perform worse than those of w/LargeBlur or DF 2.5~4.5. These ablation studies show the effectiveness of image downscaling after blurring for NERDS-raw.

**Ablation Study for Denoiser Training (NERDS+D).** Table 13 and 14 present ablation studies on blurring strengths and downscaling factors for denoiser training on the SIDD validation. This experiment uses SIDD validation to visualize the effectiveness of downscaling and blurring to noisy images. The setting of w/LargeBlur uses two times larger blur kernels than NERDS+D. Results show that our settings for NERDS+D perform the best accuracy at the diverse DFs and blurring strengths. The setting of w/oBlur performs poor PSNR/SSIM given that the downscaled images still constrain severe noise as demonstrated in Figure 8(c). Instead, the comparable results between w/Blur and w/LargeBlur indicate that the denoiser training is robust to blurring strengths. The settings for diverse DFs perform similar results given that blurring transforms noise from the P-G distribution. The transformed noise can be regarded as high-frequency textures that allow denoisers training for P-G noise.

## A.5 ADDITIONAL QUALITATIVE RESULTS

**Synthesized Noisy-Clean Image Pairs.** Figure 9 visualizes noise synthesis results using NERDS. NERDS-clean contains low-level noise while NERDS-noisy presents severe noise similar to the real noisy images. We have empirically found that denoisers *do not* learn to remove noise in NERDS-clean. This phenomenon is because the blurring and downscaling process in NERDS distorts the noise of raw images while denoisers learn to remove Poisson-Gaussian noise in the raw images. Moreover, given that NERDS does not require any clean data, NERDS successfully synthesizes noisy-clean image pairs on DND and MIT-Adobe FiveK datasets which are not available to access clean images.

**Denoised Images.** We present more images denoised by NERDS+D on SIDD (Figure 10), DND (Figure 11), and MIT-Adove FiveK (Figure 12). While AP-BSN (33), FBI-D (10), SCUNet (54), C2N (26)+DIDN (50), and CycleISP (53) often fail to restore image patterns in GT, NERDS+D successfully recover the original structures. AP-BSN uses pixel shuffle for self-supervised learning to decorrelate spatially but also loses spatial information for denoising. FBI-D learns raw image denoising for Poisson-Gaussian noise, but denoising raw images is an indirect approach for

the human visual system (or RGB space). SCUNet over-smooth or over-sharpen images. This restoration style helps to generate readable characters, but it can distort the patterns in GT (See Figure 10). C2N+DIDN is the denoiser trained by generating noisy images from clean images using both noisy and clean images on SIDD datasets. For DND benchmarks, C2N+DIDN sometimes generates artifacts or noise that significantly drops the restoration accuracy (See Figure 11). In contrast, NERDS+D uses only noisy images on DND benchmarks to generate noisy-clean image pairs for accurate denoiser training. CycleISP is the denoiser trained by paired noisy-clean images on SIDD datasets and paired images synthesized by the metadata of DND benchmarks. CycleISP performs the highest PSNR and SSIM on both SIDD and DND benchmarks compared to the methods in Table 2. However, CycleISP fails to remove the noise on MIT-Adobe FiveK (See Figure 12). This failure comes from the noise distribution of RGB images mismatched between MIT-Adobe FiveK and SIDD (or DND) datasets.

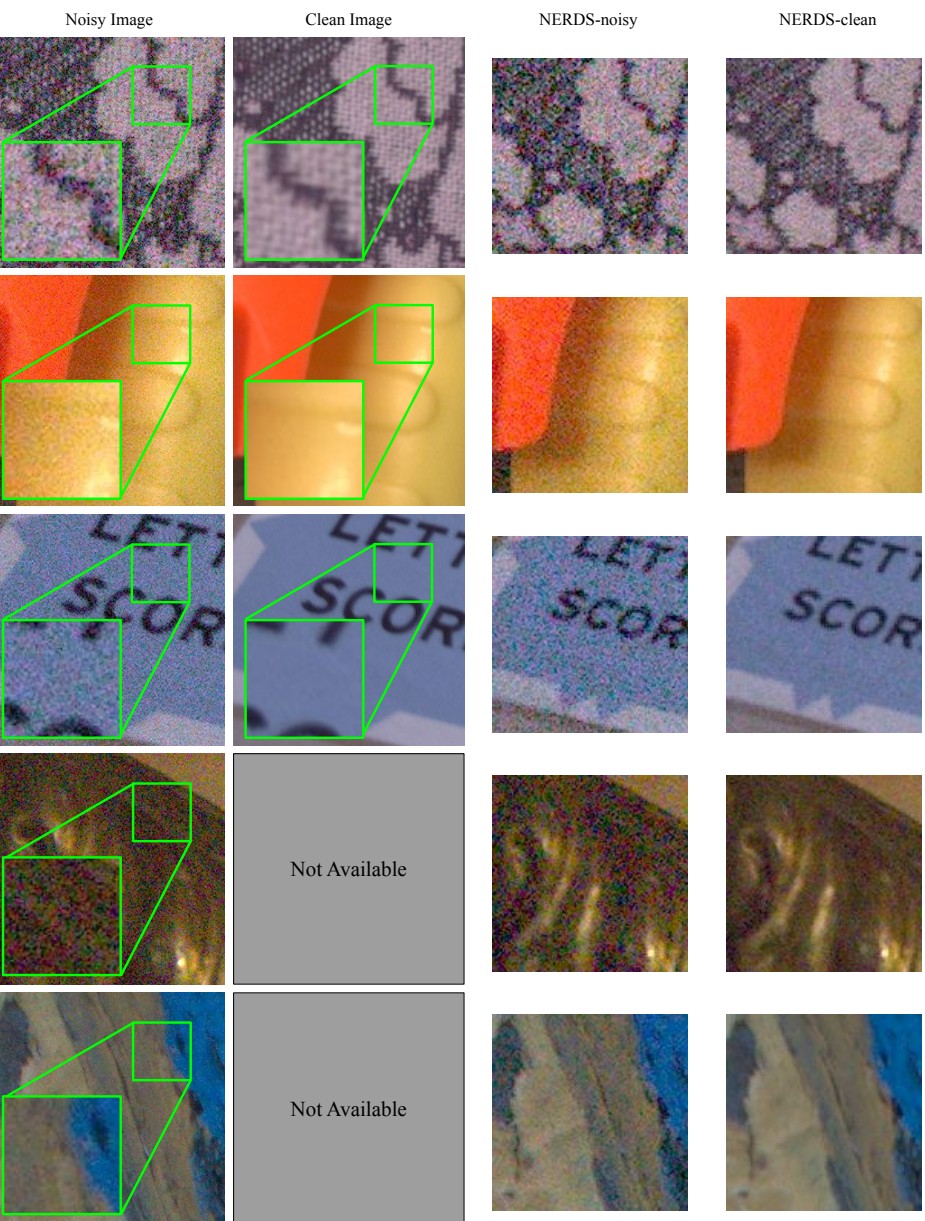

**Figure 9:** Examples of noisy synthesis results using NERDS. We upscale NERDS-noisy, NERDS-clean, and the green boxes with the scaling factor of 2 for visualization.

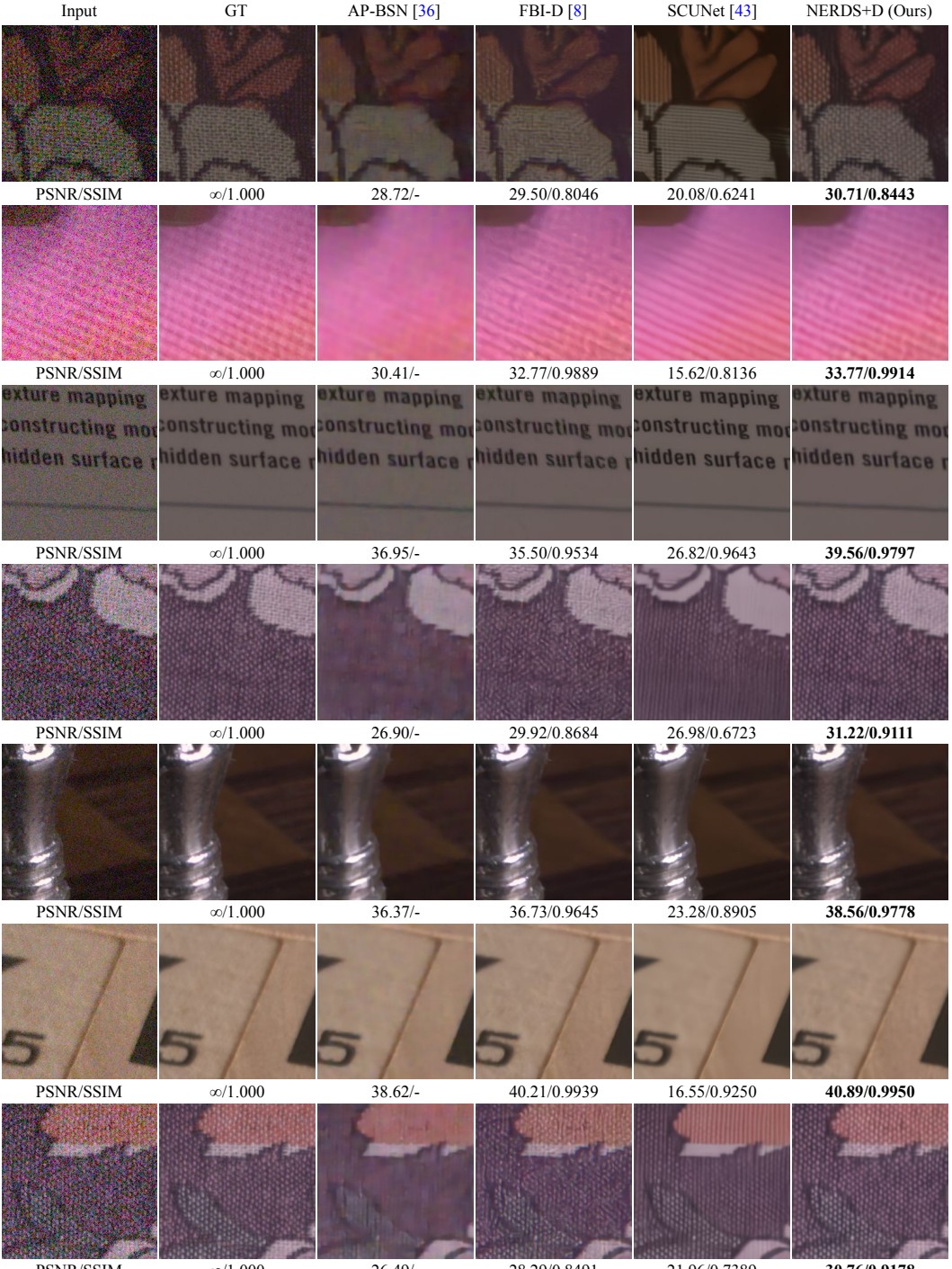

**Figure 10:** Qualitative results and PSNR (dB)/SSIM on SIDD validation.

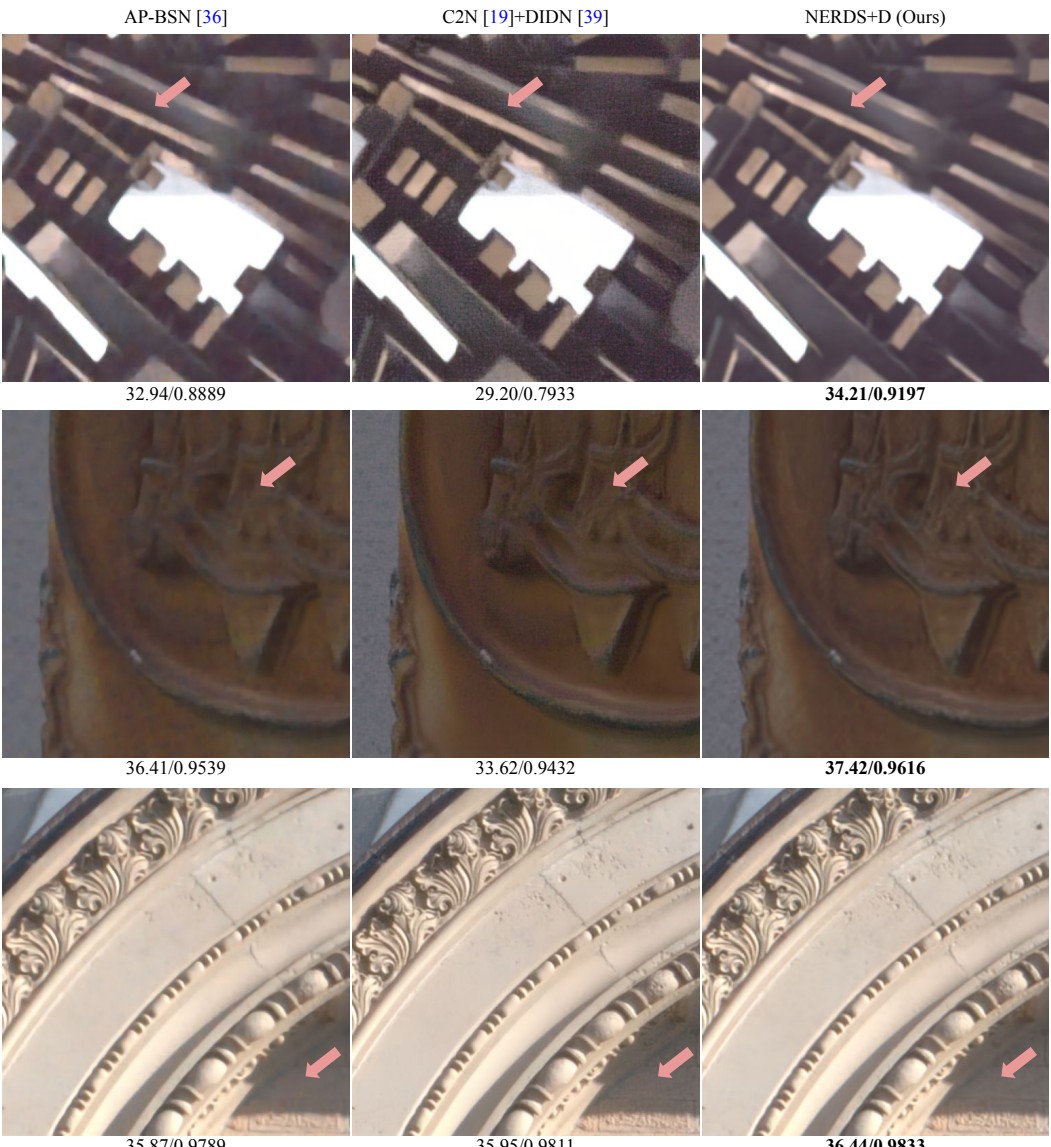

**Figure 11:** Qualitative results and PSNR (dB)/SSIM on DND.

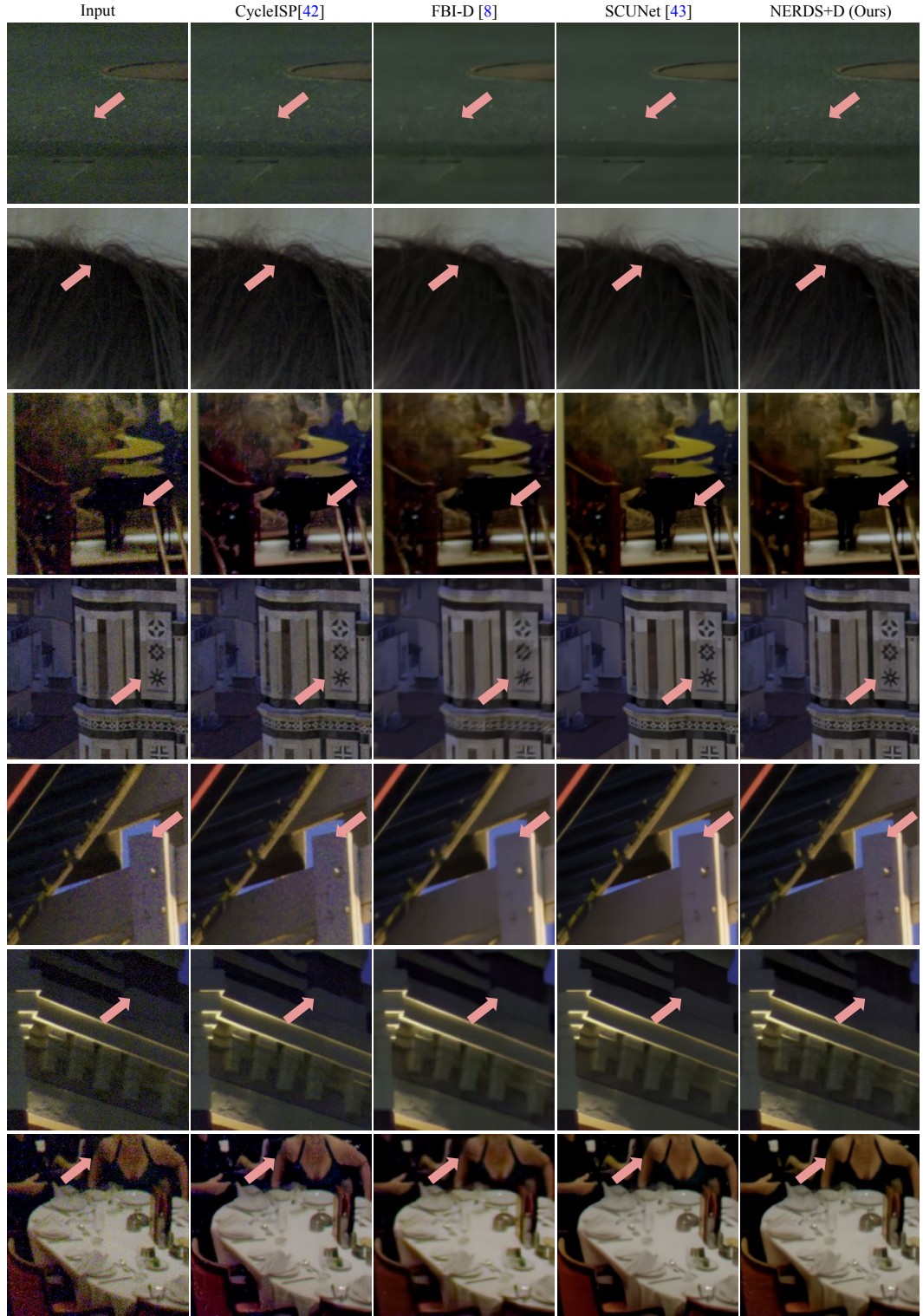

| Input | CycleISP[42] | FBI-D [8] | SCUNet [43] | NERDS+D (Ours) |

**Figure 12:** Qualitative results on MIT-Adobe FiveK. CycleISP often fails to remove noise, while FBI-D and SCUNet over-smooth or over-sharpen images. Note that NERDS+D uses only 5 test noisy images and estimated noise parameters from them for training. (*Zooming-in for the best view.*) We use $T$ of NERDS to convert raw images denoised by FBI-D to RGB images.

