# OpenReview forum: "NERDS: A General Framework to Train Camera Denoisers from Raw-RGB Noisy Image Pairs"
_ICLR.cc/2023/Conference — ICLR 2023 poster_

### Official Review · Reviewer_nCCv · 2022-10-15

**Confidence:** 5
**Correctness:** 2
**Technical Novelty And Significance:** 2
**Empirical Novelty And Significance:** 2
**Recommendation:** 3

**Clarity, Quality, Novelty And Reproducibility:**

There are many concerns in the novelty of the proposed method. Please see the weaknesses above.

**Strength And Weaknesses:**

Strength:
The provided experimental results show it outperforms better than other methods.

Weaknesses:
1. Even though Table 1 shows the proposed method can estimate noise parameters accurately and Figure 7 shows downscaling with blurring can make the variance difference between clean and noisy image closer, there lacks theoretical proof why the proposed method can accurately estimate noise parameters and analyzing the possible deviations in the proposed method. According to Figure 7, the variance of noisy images is still higher than clean ones. In this way, the estimated noise parameters based on Eq. 3 should be smaller.
2. In Figure 7, why will downsampling the clean image without blurring not change the variance of the image obviously? What is the downsampling method used? Why will downsampling 2 times have a smaller variance in (a) and (c)?
3. It is not clear whether the noise parameters are estimated for every image or for the whole dataset. According to my understanding, the proposed method estimates fixed Gaussian and Poisson parameters for the whole dataset. This is not correct. Since these parameters depend on the sensor gain or ISO. In this way, the experiment in Table 1 seems unfair. Since other methods estimate noise level image by image while the proposed one estimates it according to the whole dataset.
4. Also, using reparameterization to estimate noise parameters may be too complicated. Why not directly search all possible Gaussian and Poisson parameters and find the settings to minimize Eq. 3?
5. It seems the blur kernel and downsampling rate are important for the noise level estimation. Can the authors demonstrate how the proposed method selects the settings? More experiments are needed.
6. For isp estimation, the proposed method seems to use spatially-invariant linear operators. It may be not that accurate. In the camera isp, the processing is spatially variant e.g. lens shading, local tone mapping, contrast enhancement, etc. In addition, the number of parameters in camera isp is much larger than 3 which is the number of style parameters in the proposed method. I want to know the accuracy of the generated rgb images compared with the ground truth.

**Summary Of The Paper:**

This paper proposes a method for real image denoising. Specifically, it firstly estimates Gaussian and Poisson noise parameters. Then, it learns a style encoder to map a raw image into rgb one. In this way, it can synthesize noisy-clean image pairs to train the denoising network. The provided experiments show the proposed method performs better than existing methods.

**Summary Of The Review:**

Even though the provided experimental results are better than other methods, there exist plenty of concerns regarding noise level estimation as well as isp estimation. The authors should address them completely.

---

> ### Author Response · Authors · 2022-11-14
> **Response to Reviewer nCCv**
>
> We appreciate valuable comments from Reviewer nCCV.
> The comments indicate where readers may misunderstand our method.
> We hope that this rebuttal will correct the misunderstanding of Reviewer nCCV.
>
>
> **1. Unit of noise parameters**
>
> As the reviewer noted, each raw image can have different noise parameters depending on the camera settings.
> Correspondingly, NERDS estimates noise parameters for each raw image, denoted by $x$ in the manuscript.
> Table 1 presents a fair comparison, as it shows the average of the noise parameters estimated from each image.
>
>
> **2. Direct search for noise estimation**
>
> The direct search suggested by the reviewer cannot be used for noise estimation, given that the direct search finds infinite optimal solutions for Eq. 3 ($|Var(x) - Var(x^d+N)| = 0$).
> As shown in Figure 7, downscaling reduces the pixel variance ($Var(x) > Var(x^d)$) in general, while non-zero values of $\beta_1$ and $\beta_2$ increase the value of $Var(x^d +N)$.
> Therefore, for any $\beta_1$ satisfying $Var(x) > Var(x^d +N)$ with $\beta_2 =0$, we can find $\beta_2$ satisfying $Var(x) = Var(x^d+N)$ with the given $\beta_1$.
>
>
> **3. Theoretical background for noise estimation in NERDS (NERDS-raw)**
>
> To identify the Poisson-Gaussian (P-G) noise parameters for a raw image, NERDS optimizes Eq. 3 through gradient descent with multiple image patches.
> P-G noise has two characteristics: signal dependency and the same noise parameters for a raw image.
> Given that a natural scene usually contains various contents (or signals), each image patch has a different solution set of noise parameters for the direct search.
> Using a gradient-descent-based optimizer (e.g., ADAM), NERDS finds a unique solution for a raw image that minimizes Eq. 3 for randomly selected image patches (Section 5.1 Optimization).
>
> As the reviewer noted, the pixel variances of downscaled noisy images and original clean images shown in Figure 7 are different.
> However, the noise estimation error does not occur as much as the difference of pixel variances.
> Our noise parameter optimization through multiple image patches mitigates noise estimation errors from downscaled images.
> The properties of P-G noise discussed earlier allow downscaling to reduce the pixel variances differently between image patches.
> A well-designed optimizer statistically learns the relationship between the pixel values and the pixel variances through downscaling and finds accurate P-G noise parameters even for downscaled images with remaining noise.
> NERDS-raw uses a wide range of downscaling factors and blurring strengths to augment downscaled images for statistical learning.
> As discussed above, signal-dependent noise with various textures enables accurate noise estimation with augmented downscaled images.
>
> **4. Downscaling w/o blurring**
>
> We use bicubic interpolation (torch.nn.functional.interpolate in PyTorch) as image downscaling (Section 5.1 Downscaling).
> Given that theoretical bicubic interpolation does not perform image denoising, the similar variances through downscaling in Figure 7 are reasonable.
> We observed that the variances change for integer scaling factors.
> This phenomenon could be due to PyTorch implementation (e.g., kernel size or sampling strategy).
>
>
>
> **5. Ablation study for downscaling factor and blurring strength on noise estimation**
>
>
> We empirically determined the ranges of downscaling factors and blurring strengths that generate visually looking good images (See NERDS-clean in Figure 5).
>
> We present ablation studies on blurring strengths and downscaling factors for noise estimation in our response 2. to Reviewer aUMK.
> The results show that blurring and downscaling are important elements of NERDS-raw, but the performance of NERDS-raw is not sensitive to their hyper-parameters (blurring strengths and downscaling factors).
> For large downscaling factors and blur kernels, the noise estimation performances decrease slightly, but it still shows better performance than the methods compared in Table 1.
> Instead, the settings of w/o blur and downscaling factor (DF) of 1 estimate inaccurate noise parameters.
>
>
> **6. ISP estimation performances**
>
> NERDS estimates the RAW2RGB conversion accurately, even for spatially varying styles by patch-wise inference, where NERDS uses ISP estimation only for denoiser training with 128x128 image patches.
> Given that there is no ground truth for RAW2RGB conversion at low-resolution to measure PSNR and SSIM, we visualize generated RGB noisy/clean images (NERDS-noisy/NERDS-clean) in Figure 5, 9, and 10.
> We present additional experiments for ISP estimation performances in response 3. of Reviewer 2oSJ.

---

### Official Review · Reviewer_2oSJ · 2022-10-23

**Confidence:** 4
**Correctness:** 3
**Technical Novelty And Significance:** 3
**Empirical Novelty And Significance:** 4
**Recommendation:** 8

**Clarity, Quality, Novelty And Reproducibility:**

Generally the paper is clearly written and of good quality.  There are some small issues listed below.

The main originality comes from the noise modelling approach through downsampling and noise model fitting.  This appears to be a good advance although the reviewer has some concerns about remaining noise articulated above.

It wasn’t clear if code will be released should the paper be accepted.  Perhaps the author rebuttal could comment on this.

Smaller issues:

•	Figure 1 caption, extra period, please remove

•	Page 2, please replace “an other” to “another”


•	Page 2, footnote, please replace “Sonny” with “Sony”

•	Page 3, please replace “RELATED WORKS” with “RELATED WORK” as work is already plural


•	Page 2, please replace “test noises” with “test noise” as noise is not countable.  This issue is also on Page 4, “unseen noises” with “unseen noise”; Page 6 “learning noises” with “learning noise”

•	Page 4, it’s true modern smartphones provide multiple image styles; note however these can be applied even without user edits.  The camera will recognise the scene (e.g. beach) and optimise the ISP for that scene.  Therefore the qualifier “with user edits” is not needed.


•	Why are the results so poor for SCUNet in Table 2?


**Strength And Weaknesses:**

Denoising of images is a fundamental problem in computer vision and has seen considerable attention in the last decade particularly with the advent of deep learning.  However, as argued by the paper, supervised methods require paired noisy/clean data which is difficult to obtain.  The paper addresses this problem in an appealing way by only requiring noisy images – which are easily acquired.  A clean counterpart isn’t needed, which greatly simplifies data collection for training a denoiser.  This approach is used to synthesize noisy RGB images.

Strengths:

1.	The paper’s approach that only requires noisy RGB images as input.  This is appealing as capturing paired data often involves tedious work.  The method is able to produce noisy RGB images for RGB denoisers.  This reviewer appreciates the approach of trying to generate pairs from noisy images.

2.	The reparametrization trick to estimate the noise parameters, while simple, was interesting and something new for this reviewer.  This could be relevant for other researchers working in noise modelling and estimation.


3.	The modelling of the ISP, whilst not entirely new, was useful in this paper to produce noisy RGB images for training an RGB denoiser.  The disentangling of style was appreciated by this reviewer.

4.	The experimental results compare the method to a variety of recent methods, with the method producing convincing results.

Weaknesses:

1.	This reviewer is concerned about the downsampling technique to create “clean” images.  Clearly the images aren’t clean – for example in Figure 1, the “clean” image in part (d) is much noisier than the clean image in part (a) of the figure.  So there is remaining noise after downsampling.  The paper doesn’t really discuss this issue, although the supplementary material provides some context.  Downsampling can’t remove the image noise because it applies a low pass filter (averaging filter) to the data.  So the noise is reduced through the averaging, but not removed.  Therefore, this reviewer would prefer if the in the main paper, instead of calling it a “clean” image, the downsampled image was called a “pseudo-clean” image as done in the supplementary.  More importantly, it would be helpful if the paper could comment on the impact of this remaining noise in the downsampled image – does it impact the results negatively?  Would downsampling by a factor of 4 help more than by a factor of 2?  This reviewer was somewhat surprised to see the results in Table 1, it does appear NERDS can produce a good fit despite the remaining noise.  Perhaps the author rebuttal could comment on this point.

2.	Noise is harder to remove in the RGB domain than the RAW domain as the statistics are simpler in the RAW domain before the noise becomes more spatially correlated due to demosacing and other operations.  Why develop a denoiser to run in the RGB domain? Conceivably the method could be used to denoise in the RAW domain.  Relatedly, it could be interesting to compare the approach to the Brooks et al. paper from CVPR 2019.


3.	The reviewer is also worried about the domain gap between a “black box” ISP and the ISP that is used in the paper.  Clearly there may be a domain gap as the ISP used in the paper is only an approximation to a “black box” ISP.  Does this domain gap cause any limitations in the method?  Perhaps the proposed method does not generalize well to some phone models.

4.	Why call it NERDS?  Is this an acronym for something?


**Summary Of The Paper:**

This paper presents a novel method for training image denoising techniques, called NERDS.  The paper makes the observation that a noisy image can be downsampled to produce a pseudo-clean image, as the downsampling operation removes noise.  Based on the noise statistics between the psudeo-clean image and the original image, a noise model can be estimated in a learnable way using a reparametrization trick.  The paper also models the “black box” of an ISP disentangling style, and thereby is able to generate large amounts noisy/pseudo-clean pairs using the model and data augmentation.  These pairs are then used for training denoising methods like DnCNN.  Experimental results show the method outperforms state of the art methods.

**Summary Of The Review:**

In summary, this paper is addressing a fundamental computer vision problem of removing noise from captured images.  It brings some new ideas in terms of how to model the noise from a noisy image.  Whilst the formulation might not be perfect, it provides a practical approximate way to estimate the noise which is appealing as it can be used to generate noisy/clean pairs for image denoising.  The results are appealing as the method outperforms papers recently appearing in the literature.  Overall I think the paper has merit but it’s let down a bit by overstating assumptions and remaining gaps (e.g. residual noise in the “clean” image, differences between real and “blackbox” ISPs).  Still this reviewer believes there are some interesting ideas here.

---

> ### Author Response · Authors · 2022-11-14
> **Response to Reviewer 2oSJ**
>
> We appreciate Reviewer 2oSJ for finding our idea to be interesting and providing valuable comments.
> We wish our answers below will alleviate all the reviewer's concerns.
>
> **1. Remaining noise in pseudo-clean images**
>
> Following the reviewer's recommendation, we revised the term for downscaled raw images from ‘clean image’ to ‘pseudo-clean image’.
>
> Remaining noise in downscaled images affects both noise estimation and denoiser training.
> We present ablation studies on blurring strengths and downscaling factors for both tasks in our response 2. to Reviewer aUMK.
>
> The results firstly indicate that large blur kernels and high downscaling factors slightly degrade the performances of noise estimation and denoiser training.
> This phenomenon is because downscaling not only reduces noise levels but also removes image detail.
> Second, downscaled images without blurring (w/o blur) fail to perform accurate noise estimation and denoiser training.
> This result shows the effectiveness of noise reduction by averaging (or blurring).
> Lastly, blurred images without downscaling (downscaling factor of 1) enable training comparably accurate denoisers but estimate inaccurate noise parameters.
> The performance differences are because the remaining noise in the blurred images can be regarded as image textures by not following the P-G distribution. In contrast, the blurred images have similar pixel variances with noisy images, as visualized in Figure 7, causing severe optimization errors in Eq. 3.
>
>
> **2. Raw image denoising vs. RGB image denoising**
>
> Noise in raw images is more straightforward than in RGB images, as the reviewer commented, but high PSNR in raw space does not guarantee high PSNR in RGB space.
> This phenomenon has been presented in the Brooks et al. paper from CVPR 2019 (UPI) and our experiment in P14L042.
> Moreover, image retouching transforms image styles, where each style requires different optimal denoisers.
> UPI partially inspires NERDS, where we aim to develop a general framework by not using metadata, predetermined ISPs, and clean images.
> PSNR/SSIM comparisons between UPI and NERDS+D are 40.35/0.964 vs. 39.34/0.950 on RGB images from DND.
>
>
> **3. ISP estimation performances**
>
> NERDS estimates the RAW2RGB conversion accurately and successfully generates RGB pseudo-noisy and pseudo-clean images (NERDS-noisy and NERDS-clean) as visualized in Figure 5, 9, and 10.
> While the downscaled images are the images of interest, there is no ground truth for RAW2RGB conversion at low-resolution to measure PSNR and SSIM.
>
> To alleviate the reviewer’s concerns, we present additional experiments for ISP estimation performances in the below table.
> On SIDD validation, (noisy2noisy) denotes the transformation from raw noisy images to RGB noisy images, while (clean2clean) indicates the transformation from raw clean images to RGB clean images.
> On MIT-Adobe Fivek, (image) transforms the entire image with a single style by inferencing $T$ and $E$ once.
> Instead, (patch) transforms each part of the image with patch-wise styles by inferencing $T$ and $E$ per patch.
>
> |ISP estimation performance | NERDS|
> |:-:|:-:|
> | SIDD (clean2clean) | 48.57/0.991 |
> | SIDD (noisy2noisy) | 42.64/0.977 |
> | MIT-Adobe FiveK (image) | 39.43/0.984 |
> | MIT-Adobe FiveK (patch) | 42.40/0.949 |
>
> All results achieve high scores on PSNR/SSIM which present small domain gaps between real RGB images and synthesized RGB images.
> On MIT-Adobe FiveK, patch-wise inferences achieve better PSNR than a single inference for the entire image (42.40 dB vs. 39.42 dB), given that our ISP estimation can successfully approximate spatially-varying styles by adapting styles for small image patches (e.g., 128$\times$128).
>
> **4. Full name of NERDS**
>
> NERDS is an acronym for Noise Estimation for RGB Denoising & Synthesis.
> We revised the manuscript to include the full name.
>
> **5. Code**
>
> We will make the code publicly available for reproducibility if this paper is accepted.
>
> **6. Typos**
>
> We revised the manuscript for all commented typos and unnecessary expressions.
>
> **7. SCUNet in Table 2**
>
> SCUNet models real noise with predetermined ISPs and noise parameters and generates noisy images from high-quality clean images.
> Even though the noise modeling is realistic, it can not cover all noise distributions in the real world.
> As a result, SCUNet often generates over-smooth image contents, as visualized in Figure 6 and 12, or high-quality textures dissimilar to GT on SIDD, as presented in Figure 10.

---

> > ### Comment · Reviewer_2oSJ · 2022-11-19
> > **Thank you for the carefully prepared rebuttal**
> >
> > Thank you for the carefully prepared rebuttal, which addresses my primary concerns.  I think this paper has interesting ideas.  Reviewer's nCCv's point about providing a theoretical explanation is also interesting, however, empirically the method is shown to work.  Perhaps a theoretical argument could be provided in future work.  Nonetheless, at this stage I think paper has value and would welcome the paper in the ICLR programme if accepted to the conference.  I have increased my reviewing score to accept.

---

> > > ### Author Response · Authors · 2022-11-19
> > > **Re: Thank you for the carefully prepared rebuttal**
> > >
> > > Thank you for considering this paper worthy of publication in ICLR 2023.
> > > We will revise this paper for the camera-ready version to include empirical and theoretical explanations about our noise estimation method, as discussed in the rebuttal.
> > > Following your suggestion, we are happy to research the theoretical arguments for future work.

---

### Official Review · Reviewer_aUMk · 2022-10-24

**Confidence:** 4
**Correctness:** 3
**Technical Novelty And Significance:** 3
**Empirical Novelty And Significance:** 3
**Recommendation:** 8

**Clarity, Quality, Novelty And Reproducibility:**

The reviewer would like to appreciate the quality and novelty of this paper.

On clarity and reproducibility:
- See my comments in the previous question on optimization / model training procedure.
- One question that remains is the modern ISPs may contain denoising block. As the RAW2RGB process is learned from real noisy raw/RGB image pairs, it is possible that some degree of denoising is already learned implicitly in T, which potentially may impact the denoising strength of D.
- Generalizability: please clarify if at test time, D needs to be fine-tuned / retrained for device specific noise levels and ISPs. Based on Sec A.1.3 it looks like this is required and time-consuming to get the best denoising performance, meaning that we will also need to access the noisy RGB/raw image pair at test time training. If so, please clarify if the comparison in table 2 was done fairly, i.e. did the other methods have access to training set at test time?




**Strength And Weaknesses:**

Strength:
- Creative way of synthesizing raw noisy/clean image pairs based on the observation that true image texture is more robust to pixel variance through downscaling than real noises.
- Poses the ISP estimation as a style encoding and transfer problem. This, in combination with the innovation above, greatly relax the dataset requirements on real noisy/clean image pairs, ISPs, metadata etc. which are not always available or accurate especially at test time.
- While key proposals such as pseudo clean image generation via downsampling may seem to be empirical, the authors made an attempt to justify this with data. The ablation study also seems adequate.
- NERDS outperforms previous works with simpler denoiser network, demonstrating the effectiveness of the noisy/clean image pair synthesis, style disentanglement-based ISP estimation, as well as the rich data augmentation.

Weaknesses:
- The work still relies on raw/RGB noisy image pairs for training, although only RGB noisy image is required for denoising at test time. I would recommend that the authors rethink the title, "NERDS: A GENERAL FRAMEWORK TO TRAIN CAMERA DENOISERS FROM SINGLE NOISY IMAGES", as the claim on training from single noisy images may be ambiguous and potentially misleading (it actually requires both raw and RGB noisy images for training and testing ideally, thus the "s" in "images").
- While the authors have made an attempt to explain why the pseudo clean image (with residual noises) works in practice due to various observations / hypothesis, it may add more insights if ablation study can be conducted to assess the relative contribution from factors such as blurring strengths, downscaling factor, augmentation for noise parameter, scale/intensity, and style etc.
- Although it is claimed by the authors that PG parameters in metadata may not be accurate, and that NERDS-raw does a better job, the audience may still appreciate more justification on this. For example, what is the impact of inaccurate PG parameters on the overall RGB denoiser performance?
- On reproducibility, it would be helpful if the authors could clarify if the PG optimization, ISP network T training, and denoiser D training are done subsequently or jointly.
- I recommend that the authors consider referencing the supplementary material in the main paper, as the audience may walk away with many questions just by reading the main paper only.

**Summary Of The Paper:**

In this paper a learning-based camera denoising framework is introduced. Noisy raw/RGB image pairs are used for training, while the method takes a noisy RGB image only at test time for denoising. The authors designed a framework that models the raw sensor noise and ISP transformed noise separately.

Instead of relying on paired noisy/clean images for training, the authors exploited the use of image downsampling and optimization-based noise estimation and synthesis to create pseudo noisy/clean image pairs at lower resolution, purely from noisy images. For estimating the Poisson-Gaussian coefficients for the raw sensor noise, a re-parameterization trick is applied and an optimization problem is solved such that the pixel variances can match between synthesized low-res noisy image and the original full-res noisy input. While P-G parameters are often provided by camera manufacturers in metadata, they may not be accurate and the authors proved that the proposed approach leads to more accurate estimation than prior arts.

For the RAW2RGB conversion, this ISP process is posed as a style transfer problem, where the style is learned from the raw-RGB noisy image pair (at original resolution) and is used to convert the raw image to RGB.

Once the P-G model (N) and RAW2RGB conversion network T are determined, a given denoiser network can be trained on the synthesized low-res noisy/clean image pairs with a number of data augmentation tricks. The overall objective function can be seen in Eq (8).

Thorough qualitative, quantitive, as well as ablation studies have been conducted on various datasets, demonstrating the effectiveness of the framework, and justifying the design decisions.



**Summary Of The Review:**

While minor clarity issues remain, NERDS is a novel and effective framework for camera denoising. The main drawback (and overselling) though, seems to be the claim on requiring single noisy images only -- in practice, both the raw and RGB noisy images are required to get the best performance out of the denoiser. Please consider making the problem setting and contribution more concrete and transparent. Please also consider referencing some key justifications in the supplementary material from the main paper, as the readers may otherwise believe some design decisions lack sufficient justification (empirical, theoretic). As such, I recommend a borderline accept (6) for the work, and lean towards score 7 if these can be addressed in the revision.

---

> ### Author Response · Authors · 2022-11-14
> **Response to Reviewer aUMk**
>
> We thank Reviewer aUMk for the valuable comments and suggestions.
> We tried our best to answer the questions.
>
> **1. Paper title**
>
> We agree with the reviewer’s concerns about our title and amend it as follows:
>
> NERDS: A General Framework to Train Camera Denoisers from Raw-RGB Noisy Image Pairs
>
> **2. Ablation study for noise estimation and denoiser training**
>
> Table 3 addresses an ablation study for denoiser training to present augmentation effects on noise parameters, image scale, image intensity, and style.
> Each factor improves the denoising accuracy.
> Additionally, the below tables present ablation studies on blurring strengths and downscaling factors for denoiser training on SIDD validation.
> This experiment uses SIDD validation to visualize the effectiveness of downscaling and blurring to noisy images.
> The setting of w/ large blur uses two times larger blur kernels than NERDS+D and NERDS-raw.
>
> | Downscaling factor (DF)  | 1 | 1.5~2.5 (Ours) | 2.5~4.5 |
> |:-:|:-:|:-:|:-:|
> | SIDD | 37.66/0.942 | **38.02**/**0.946**| 37.89/0.943 |
>
> | Blurring strength | w/o blur | w/ blur (Ours) | w/ large blur |
> |:-:|:-:|:-:|:-:|
> | SIDD | 26.99/0.642 | **38.02**/**0.946** | 38.01/0.944 |
>
> Results show that our settings for NERDS+D perform the best accuracy at the diverse DFs and blurring strengths.
> The setting of w/o blur performs poor PSNR/SSIM given that the downscaled images still constrain severe noise as demonstrated in Figure 7(c).
> Instead, the comparable results between w/ blur and w/ large blur indicate that the denoiser training is robust to blurring strengths.
> The settings for diverse DFs perform similar results given that blurring transforms noise from the P-G distribution.
> The transformed noise can be regarded as high-frequency textures that allow denoisers training for P-G noise.
>
> For noise estimation (NERDS-raw), we design ablation studies on blurring strengths and downscaling factors on BSD68 as follows:
>
> | Noise level $(\beta_1, \beta_2)$|  w/o blur | w/ blur (Ours) | w/ large blur |
> |:-:|:-:|:-:|:-:|
> |(0.100, 0.0200)| (0.080, 0.0144) | (**0.100**, 0.0229) | (0.119, **0.0220**) |
>
> | Noise level $(\beta_1, \beta_2)$ |  DF 1| DF 1.5 ~ 2.5 (Ours) |  DF 2.5 ~ 4.5|
> |:-:|:-:|:-:|:-:|
> |(0.100, 0.0200)| (0.065, 0.0085) | (**0.100**, **0.0229**)|(**0.100**, 0.0279) |
>
> The settings of w/ blur and DF 1.5$\sim$2.5, which indicates NERDS-raw in Table 1, present better performance than compared settings.
> The settings of w/o blur or DF 1 perform worse than those of w/ large blur or DF 2.5$\sim$4.5.
> These ablation studies show the effectiveness of image downscaling after blurring for NERDS-raw.
>
> **3. Importance of accurate P-G parameter estimation**
>
> Following the reviewer's suggestion, the below table presents an ablation study on noise parameters in SIDD validation.
> For a fair comparison, all experiments train the model of $D$ using all augmentation in Table 3.
> Metadata uses noise parameters in SIDD metadata.
> Random uniformly samples noise parameters between the minimum/maximum values of the noise parameter estimated by NERDS-raw.
>
> |Noise parameters| NERDS-raw | Metadata | Random|
> |:-:|:-:|:-:|:-:|
> SIDD | **38.51**/**0.950** | 37.92/0.941 | 37.56/0.937 |
>
> NERDS-raw has better denoising performance than all compared methods.
>
> **4. Reproducibility**
>
> Each step of our framework is done subsequently.
>
> **5. Referring to Appendix**
>
> We revised the manuscript to refer to the sections of the Appendix.
>
> **6. Denoising by ISPs**
>
> We assume that the CNN part of T can estimate the denoising block of modern ISPs regarded as a type of RAW2RGB conversion.
> Given that the denoising block is not perfect, our denoisers enable learning additional noise removal from generated pseudo-noisy and pseudo-clean image pairs.
>
> **7. Generalizability and fair comparisons**
>
> Self-supervised-learning-based methods and noise-synthesis-based methods (N2V, AP-BSN, FBI-D, and C2N+DIDN in Table 2) require test-time adaptation like NERDS.
> These methods use test noisy images for denoiser training or noise synthesis, so the comparisons are fair.
> For supervised learning, the test-time adaptation includes capturing noisy/clean image pairs.
> Compared to data collection, our noise and ISP estimations are efficient processes for time and resources.
>
> **8. Problem setting and contributions**
>
> The problem setting of this paper is learning RGB image denoisers from raw-RGB noisy image pairs.
> The major contribution of this paper is generating RGB noisy-clean image pairs at low-resolution.
> To this end, we propose a novel noise estimation method by downscaling raw noisy images and regard the downscaled images as pseudo-clean images.
> We also propose a network for ISP estimation that transforms raw noisy-clean image pairs to RGB space.
> Finally, we introduce diverse data augmentation techniques in our framework to train accurate denoisers.

---

> > ### Comment · Reviewer_aUMk · 2022-11-28
> > **Thank you for the rebuttal. Raised rating to "accept".**
> >
> > I appreciate the authors' detailed rebuttal and consideration of better reflecting the nature of NERDs in the paper title. Raised my rating from borderline accept to accept.

---

> > > ### Author Response · Authors · 2022-11-30
> > > **Re: Thank you for the rebuttal. Raised rating to "accept".**
> > >
> > > We appreciate your detailed advice and comments on the paper title, the problem status, and the experiments for a better paper.
> > > We will incorporate the issues from the discussion and clarify the manuscript for the final version.

---

### Official Review · Reviewer_HZ4f · 2022-10-24

**Confidence:** 3
**Correctness:** 3
**Technical Novelty And Significance:** 3
**Empirical Novelty And Significance:** 2
**Recommendation:** 8

**Clarity, Quality, Novelty And Reproducibility:**

Minor hiccups in argumentation exist, e.g.
P6L19 claims ‘To avoid learning identity mapping from the input y to the output \hat{y}, we first adopt image scale augmentation (SA) that changes the image resolution.’ Comparing Figure 4, there is no path from y to \hat{y} allowing to learn identity, even without SA. The next sentence then claims ‘[…] SA randomly samples the scaling factor to prevent the encoder from learning noises’. It should be written clearly, for what purpose SA has been introduced and it would be informative to see, if performance of the method really drops when removing it.

Clarity is mainly reduced due to language issues. E.g. the paper frequently violates the rule that the definite article (the) is used before a noun to indicate that the identity of the noun is known to the reader or before unique nouns. It is somewhat distracting, when this rule is not kept, as it leads the reader to consider if they overlooked a previous mentioning.
Other examples of language issues, notation:

P1L5: ‘Intuitively, downscaling easily removes high-frequency noises than natural textures.’ The word ‘more’ is missing, allowing to use comparison by ‘than’: ‘Intuitively, downscaling removes high-frequency noise more easily than natural textures.’ Same on P5L16: ‘This phenomenon indicates that the true signal z is robust to the pixel variances through downscaling than real noises.’

P1L-10: ‘The first line of works synthesizes the realistic noise from clean images […]’. It remains unclear, how noise can be synthesized from clean images, and if this is really meant. Presumably noisy images are generated, where the noise not necessarily depends on the clean image.

P1L15: ‘synthetic raw images’: no synthetic raw images are used in the paper. Only rgb images are simulated.
P5, paragraph ‘Optimization’: the notation $a \times e^y$ is unusual. Either use $ae^y$ or $a \times 10^y$ .

Novelty: The noise parameter estimation, as well as the estimation of the raw2rgb converter are new, the underlying models are not but this is also not claimed. Using down-sampling for noise reduction in order to then add noise to produce noisy/clean image pairs seems to be new. All simple but effective. However, the observation, that down-sampling increases SNR is not new and obvious in Fourier space, as high-frequency regions have lower SNR in typical images. This observation should therefore be presented less pronounced.

Reproducibility: While the method is clear overall, details of implementation are not. Especially exact network architectures remain unclear. A table, figure, or code would help.

Results in Tables 2 and 3 seem to be inconsistent, as numbers presented for NERDS+D in SIDD (with full augmentation in Table 3) are different.



**Strength And Weaknesses:**

Strengths of the paper are in the well-designed, easy to understand ingredients of the method and the improved denoising performance wrt. competitors. It is overall well-structured and presents clear experiments. It also provides a rich overview on related work.

Main weaknesses are language issues that sometimes lead to unclear train of thoughts within sentences or paragraphs (see examples below).


**Summary Of The Paper:**

The paper proposes and evaluates a training method for neural network-based image denoising when raw images and corresponding processed (e.g. demosaiced) rgb images are available. It leverages the known observation that SNR increases when spatially downscaling an image, and thus realistic low and high-noise images can be generated at lower resolution. The method consists of several carefully chosen ingredients, e.g. (1) a scheme for estimating parameters of a Poisson-Gaussian noise model on raw images, (2) a scheme for estimating parameters of a signal processing pipeline converting raw to rgb, (3) augmentations using the estimated parameters, (4) supervised training on noisy/clean low-resolution rgb image pairs simulated from raw images. The method is evaluated for two different neural networks and compared to sota methods that have not been trained or real (hard to get) noisy/clean images. In this comparison the proposed method performs preferably.

**Summary Of The Review:**

Overall a paper with merits due to its simple but novel and effective content. However, it suffers from language issues. This brings the paper well into the range of being acceptable, but keeps it from being a ‘good paper’ overall.

---

> ### Author Response · Authors · 2022-11-14
> **Response to Reviewer HZ4f**
>
> We appreciate Reviewer HZ4f for commenting that our method is simple yet novel and effective.
> Based on reviewers' comments, we will polish all text in the manuscript and the appendix to be “a good paper” by the time this paper is published.
> We clarified all the issues mentioned by Reviewer HZ4f below.
>
> **1. Scale Augmentation ($SA$) for ISP estimation**
>
> Given that image scaling transforms image content and noise distribution, $SA$ prevents the encoder from learning identity mapping and noise generation, regardless of the network architecture of $T$.
> However, as the reviewer mentioned, $T$ has no path from $\boldsymbol{y}$ to $\hat{\boldsymbol{y}}$ by adopting a bottleneck architecture.
> We revised the description of $SA$ as follows:
>
> "For accurate RAW2RGB conversion for downscaled images, which are unseen in training, we first adopt image scale augmentation ($SA$) that changes image resolution.
> $SA$ generates multiple training patches for a style (or a patch) by randomly sampled scaling factors to prevent overfitting to a few training data."
>
> The below table presents the performances of ISP estimation to check the effectiveness of $SA$.
> We use raw/RGB noisy images of SIDD validation for training and measure PSNR/SSIM between RGB clean images and raw clean images transformed by $T$ (clean2clean).
> We also report PSNR/SSIM between RGB noisy images ($\boldsymbol{y}$) and raw noisy images transformed by $T$ ($\hat{\boldsymbol{y}}$), which indicate training accuracy (noisy2noisy).
> Without $SA$, $T$ performs worse on raw clean image conversion while achieving better training accuracy.
>
> |ISP extimation performance| w/ $SA$ | w/o $SA$|
> |-|-|-|
> |Training accuracy (noisy2noisy) | 42.64/0.977 | **43.32**/**0.995**|
> |Test accuracy (clean2clean) | **48.57**/**0.991** | 47.37/0.990|
>
>
> **2. Grammar error**
>
> We revised the usage of ‘the’, ‘more’, and the notation.
> Please check our revised manuscript.
>
>
> **3. P1L10**
>
> Camera noise is generally assumed to depend on clean images (e.g., Poisson-Gaussian noise).
> However, we agree with the reviewer's comment and modify the sentence as follows:
>
> The first line of works generates realistic noisy images from clean images to utilize supervised denoiser training as visualized in Figure 1(b).
>
>
>
> **4. P1L15**
>
> ‘synthetic raw images’ indicates the raw noisy-clean image pairs at low-resolution.
> We changed ‘synthetic raw images’ to ‘downscaled raw images’ in the revised manuscript.
>
>
>
> **5. Novelty**
>
> Thanks for finding that the proposed method is novel, simple, and effective.
> To the best of our knowledge, this paper is the first attempt to generate training pairs at low-resolution without clean images or metadata for camera denoisers.
>
> As the reviewer’s comment, it is well known that downscaling increases the signal-to-noise ratio (SNR).
> However, our observation includes a new finding on measuring image quality.
> Given that SNR quantifies the image quality of an image by comparing it to its true signal, SNR cannot measure the noise level of an image in our setting; the true signal (or a clean image) is not accessible.
> For this, our observation shows that the difference of pixel variances through downscaling can be used to measure noise levels without any clean images (or true signals).
> The observation deserves emphasis because NERDS enables estimating the noise parameters based on the difference of pixel variances (Eq. 3).
>
>
>
> **6. Reproducibility**
>
> We will make the code publicly available with the implementation details (e.g., network architectures).
>
> **7. Score differences between Table 2 and 3**
>
> We use different datasets in SIDD (benchmark and validation) for Table 2 and 3.
> The benchmark/validation datasets have publicly unavailable/available clean images.
> In our opinion, the validation dataset is suitable for ablation studies for reproducibility.
> NERDS+D in Table 2 matches the models in (5) and (6) in Table 3.

---

> > ### Comment · Reviewer_HZ4f · 2022-11-25
> > **Thanks for the convincing rebuttal. Raised score to 'accept'.**
> >
> > Thank you for the well-prepared rebuttal addressing my questions satisfactorily. I think the paper is sufficiently solid and interesting to be shown at ICLR. I raised my score to ‘accept’.

---

> > > ### Author Response · Authors · 2022-11-30
> > > **Re: Thanks for the convincing rebuttal. Raised score to 'accept'.**
> > >
> > > We are grateful that you rated our paper as interesting and solid enough to be 'accepted' for ICLR.
> > > We will polish this paper further for the final version, including the issues from the discussion.

---

### Author Response · Authors · 2022-11-17
**Any questions for discussion?**


We thank again to all the reviewers.

Please let us know if the reviewers have additional questions or concerns about this paper and the authors' responses.
 We are happy to answer or discuss them.

---

### Decision · Program_Chairs · 2023-01-20

**Decision:**

Accept: poster

**Justification For Why Not Higher Score:**

The paper introduces an effective method that relies on new ideas and a variety of carefully selected old ones. However, since many elements of the method (such as the downsampling idea) are well known in the community, since the method actually relies on using image raw+RGB image pairs (which is a slight weakness in terms of the significance of the setup considered), and since the paper could be better written and presented, this is not a paper for a spotlight or an award.



**Justification For Why Not Lower Score:**

The paper provides a well-designed framework for self-supervised learning of denoisers, based on interesting ideas such as the particular synthesis of raw/clean image pairs, and shows with well-designed comparisons and ablation studies that the method works well.

**Metareview: Summary, Strengths And Weaknesses:**

The paper proposes a framework to train denoisers based on pairs of raw and RGB images. The method only relies on noisy images. The paper shows that the method enables training CNN-based denoisers well, relative to training with other self-supervised losses that do not rely on having seen clean images for training.

Strength:
The paper provides a well-designed framework for self-supervised learning of denoisers, based on interesting ideas such as the particular synthesis of raw/clean image pairs.


Weaknesses:
- Some language issues, concrete examples are given by R1. Most have been fixed.
- Both raw and RGB noisy image pairs are required at training, while only noisy RGB images are used at inference. This was initially not made sufficiently clear but has been clarified better in the revised version. I recommend to also explain early on in the paper what raw and RGB noisy image pairs are, and how they are obtained in practice, this is not sufficiently clear in the revised version.
- The paper does not provide a proof stating under which assumptions the estimation of the noise parameters (e.g., the variance) is sound. Adding such as proof would add value to the paper, since it would illustrate under what assumptions the method is sound and when it is not.



**Note From Pc:**

if the above contains the word "oral" or "spotlight" please see: "oral" presentation means -> notable-top-5% and "spotlight" means -> notable-top-25%. As stated in our emails, we are disassociating presentation type from AC recommendations

**Summary Of Ac-Reviewer Meeting:**

Here is a summary of the online discussion:

R1 notes that the method consists of several carefully chosen ingredients and that it performs well. Also notes a few concrete language issues and writes that this at times leads to unclear lines of thought. Moreover, some details of the implementation are not given. The authors addressed those issues, and the reviewer raised their score from 6 to 8.

R2 notes that NERS is an effective method and thorough comparisons and ablation studies have been carried out. R2 points out that there are language issues, and that it is not sufficiently clear that raw + RGB pairs are required at training. The authors addressed those comments changed the paper's title, and the reviewer raised their score from 6 to 8.

R3: The reviewer recommends accepting (8) after the rebuttal, based on the paper providing a practical way to estimate the noise and for performing well in comparison to the state-of-the-art.

R4 also appreciates that the method outperforms competitors. However, R4 recommends rejecting the paper for a lack of theory and other concerns regarding noise level estimation. The authors addressed the concerns, and I think that while theory would be nice, this is not necessary since the authors demonstrate through thorough experiments and ablation studies that the method works well.